# The Entropy of Entropy: Are We Talking about the Same Thing?

**DOI:** 10.3390/e25091288

**Published:** 2023-09-01

**Authors:** Søren Nors Nielsen, Felix Müller

**Affiliations:** 1Department of Chemistry and Bioscience, Section for Bioscience and Engineering, Sustainable Bioresource Technology, Aalborg University, A.C. Meyers Vænge 15, DK-2450 Copenhagen, Denmark; 2Department of Ecosystem Management, Institute for Natural Resource Conservation, Christian-Albrechts-Universität zu Kiel, Olshausenstrasse 75, D-24118 Kiel, Germany; fmueller@ecology.uni-kiel.de

**Keywords:** biology, ecology, hierarchy, maximum entropy production, minimum entropy production, thermodynamics

## Abstract

In the last few decades, the number of published papers that include search terms such as thermodynamics, entropy, ecology, and ecosystems has grown rapidly. Recently, background research carried out during the development of a paper on “thermodynamics in ecology” revealed huge variation in the understanding of the meaning and the use of some of the central terms in this field—in particular, entropy. This variation seems to be based primarily on the differing educational and scientific backgrounds of the researchers responsible for contributions to this field. Secondly, some ecological subdisciplines also seem to be better suited and applicable to certain interpretations of the concept than others. The most well-known seems to be the use of the Boltzmann–Gibbs equation in the guise of the Shannon–Weaver/Wiener index when applied to the estimation of biodiversity in ecology. Thirdly, this tendency also revealed that the use of entropy-like functions could be diverted into an area of statistical and distributional analyses as opposed to real thermodynamic approaches, which explicitly aim to describe and account for the energy fluxes and dissipations in the systems. Fourthly, these different ways of usage contribute to an increased confusion in discussions about efficiency and possible telos in nature, whether at the developmental level of the organism, a population, or an entire ecosystem. All the papers, in general, suffer from a lack of clear definitions of the thermodynamic functions used, and we, therefore, recommend that future publications in this area endeavor to achieve a more precise use of language. Only by increasing such efforts it is possible to understand and resolve some of the significant and possibly misleading discussions in this area.

## 1. Introduction

A recent review of the application of thermodynamics in ecology revealed that the number of implementations that have been based on such an understanding has grown immensely over recent decades [1]. At the same time, some problematic issues appear to arise when attempting to seek scientific explicitness, accuracy, and consistency. This critical viewpoint is generally valid when looking at the coupling between the actual equations used to describe environmental problems in thermodynamic terms. The mathematical equations do not always match the semantic formulations and logic used. In the case of the entropy concept, its usage is not always unambiguous in meaning or interpretation.

This sometimes-diffusive picture may be the reason for most of the troubles that are met when applying “thermodynamics” to far-from-equilibrium conditions, such as we find in domains such as life, biology, or ecosystems. At its core, the related uncertainties can essentially be ascribed to a duality in our understanding of the entropy concept and what we assume it to express. It can be seen either as a measure of state, i.e., as an indication of the probability in the distribution of elements or as a consequence of the change in energy quality (availability) due to the irreversibility related to a specific process, or sometimes it may be used as both.

Already, Lotka [2,3] realized how complicated the situation can be in biological systems. The number of particles and hence the possible relations between them would tend to increase, resulting in a simultaneous increase in biomass and structure, which in turn would function to further increase the energy flows, inputs and outputs, through the system. In the course of evolution, these processes may have increased toward some maximum conditions. Later, Odum and Pinkerton [4] expanded on this observation in their studies on the maximum power principle. All in all, these efforts showed that the description of the thermodynamic development of biological systems is multifaceted and contains exactly the above-mentioned duality between the distributions and sizes of the constituent elements and energy flows.

The above-mentioned review [1] illustrates the complexity of this problem, in particular when different interpretations of the concept of entropy are used at various biological levels. This constellation means in a very simplified formulation that approaches claiming to deal with entropy can generally be divided into research describing *either* distributional patterns of ontological elements *or* estimations of actual dissipations. E.P. Odum [5] dealt with this ambiguity in his observation of increasing diversity—both as evenness and heterogeneity—in constituent components or particles sensu lato. This position can be seen as complementary to a view where it is actual entropy formation or energy dissipation, which is the focus of the given study.

From these phenomenological observations, causal and teleological discussions arise that address questions such as whether these systems will develop toward a state of minimum or maximum entropy? And do we talk about entropy as distribution or production or maybe even both? And will the final evolutionary state just be optimal with respect to prevailing *constraints* [6] or *restrains* as described by Bateson [7]?

The challenge in ecological theory is now not only to distinguish between various types of usage, which is most likely to depend on the level of hierarchy and the corresponding particles to which the concept is applied, but also to identify the important linkages between the two ways of using the concepts, if such a connection exists?

In the present paper, we will attempt to shed light on the origins of the many discussions that have taken place in this area. As mentioned above, the background work of an earlier paper revealed that there might be some underlying order in the confusion about how to apply and interpret thermodynamics in general and the entropy concept in particular, when the systems under consideration are part of any biological domain. Many reasons point back toward inconclusive discussions in physics regarding the extension of the validity of thermodynamic laws to apply to far-from-equilibrium conditions. Another set of issues seems to focus on entropy as a measure of the probability of distribution as opposed to a measure of the dissipation of energies as such, i.e., the continuous degradation of energy quality that occurs during processes. All in all, there has been an ever-increasing need for precision in not only the semantics but also strict physical definitions when working in this field. It is our hope that this paper will raise awareness regarding confusing usages in the past and at the same time serve to encourage greater precision in future formulations and discussions. Therefore, we will follow the following main questions within this paper:(1)Why do we have such a strong variety of comprehensions of entropy in our different sciences?(2)What are the basic starting points of the terminological diffusion around the term entropy?(3)How can we avoid general comprehension problems when analyzing entropy in systems far from thermodynamic equilibrium?(4)How can we integrate the hierarchical organization of many open systems into the systems-based analysis and indication of entropy?(5)How can the different extremum principles in entropy analysis concepts be distinguished?(6)Is it possible to integrate the ideas of entropy maximization and minimization?

The discussions about those topics will start with a description of some essential background(s) in Section 2 and a depiction of the leading entropy controversies (Section 3). Thereafter, the concept of ontic openness will be introduced (Section 4), and the resulting problems in connecting classical thermodynamic viewpoints to far-from-equilibrium (FFE) conditions will be illuminated. In the Section 6, we discuss thermodynamics in biological hierarchies and analyze entropy conditions at different levels of the biological hierarchy (Section 7). The forthcoming parts of this paper include suggestions and consequences, starting with demands for new interpretations (Section 8), proposing recommendations for future work (Section 9), and suggesting some ideas, which may facilitate the demanded steps (Section 10).

## 2. Some Essential Background(s)

The implementation of thermodynamic principles, such as entropy and exergy, in order to improve our understanding of the functionality, efficiency, adaptation, evolution, development, variation, and selection in both biological systems in general and ecosystems specifically has been advocated for several decades now. If we accept Lotka’s seminal papers from 1922 [2,3] and take his presentations on this issue as a starting point, we can view his statements about organisms’ *competition for free energy* and *maximization of power* as some of the first examples of the application of a thermodynamic interpretation to biological systems [8]. Thus, while working on this paper, we are in fact celebrating a 100-year anniversary of this topic in the biological sciences.

Over the years, applications have occurred at almost all levels of the biological hierarchy resulting in many interesting but also, from time to time, apparently conflicting statements and findings. Unfortunately, the results and conclusions from analyses at various levels have been based on very different systems that use many different approaches to thermodynamics. The area thus ranges from mere energetic, first-law analysis, to increasingly advanced methods based on the second law. Here, we find the applications of concepts, such as entropy and exergy, and other ideas emerging from differing world views.

Energetic ways of analysis have been popular among biologists with a physiological and autecological orientation and, in most cases, concentrate on mapping how energy is taken up and invested by organisms. Odum’s and Pinkerton’s [4] interpretation of Lotka’s *maximum power principle* [2] brings us close to the border between the two laws and thus to a possible interface (e.g., [9]). With the introduction of the second law, which follows, among others, the ideas of Lotka around competition for free energy as sketched above, but also reflects the concepts of Schrödinger [10], Brillouin [11,12], Odum [13], Prigogine [14,15], Jørgensen [16], Ulanowicz and Hannon [17] Schneider and Kay, [18], Weber and Depew [19], and many others, it is now clear that the development and evolution in biological systems are not only dependent on energy quantity; but also *energy form, it’s quality, and thus the formation of entropy*; i.e., dissipation. That is, energy quality is of maybe even greater importance than just mere energy.

Meanwhile, where the application of the first law from a backward perspective appears to have been almost trivial, this can hardly be said about the transfer of second law approaches, such as entropy, to biological systems in a broader sense. The problems one meets when working in the area arise from issues that are deeply rooted in physics. Here, we deal with the fact that entropy, strictly speaking, only relates to ideal gases and systems at near-equilibrium conditions. Such conditions are hard to meet since all biological systems are operating under far-from-equilibrium (FFE) conditions [20]. Furthermore, Prigogine and coworkers’ expansion of thermodynamics to FFE conditions based on Onsager’s work [21,22] continues to assume a linear relationship between forces and fluxes [23], which actually is not the case [24].

In brief, the implementation of entropy in biology brings as an implicit consequence that one works with a concept that has only a vague definition, if any at all, under the conditions normal to biological systems [14]. This puts an extra demand on authors in the area to explain their use of these terms with stringency and consistency.

The basic equations used for entropy, the Boltzmann equation [25] or its extended version, the so-called Boltzmann–Gibbs equation [26], have been widely applied at almost all levels of the biological hierarchy. One problem now becomes clear, namely that the configurational description of the various systems in terms of their respective ontological particles, which the equations are now used for, is far from the original description of possible microstates in a gas that consists merely of atoms or molecules. It cannot be treated as such, although some authors seemingly accept the metaphor [27]. At almost all levels of the biological hierarchy, the ontological units or particles used in such entropy calculations differ greatly between levels that are often referred to as *levels of organization* [28]. This raises some questions that need to be answered. When we shift between hierarchical levels, we, at the same time, accept a shift in the ontological units used in calculations or, as we will call them here, “particles”, which come to differ between the various levels under consideration. But are we then really talking about classical, conventional entropy [29]? Or is it “something” else? In the latter case, this “something” needs to be defined. And, in the end, are we just observing systems with similar behavior, which may present an analogous way of understanding without being the “real”, classical entropy of physics any longer? And how may observations on such systems be translated into thermodynamics and entropy? Or are we dealing with entropy at all? If not, how do our observations then relate to thermodynamics? A recent review [1] identified some of these different uses of entropy and, in addition, a particular use of entropy that appears to depend on the background of the authors or the area of implementation; for example, biodiversity studies or comparisons of landscapes [1].

Another problem relating to the fundamental physical understanding of the systems is whether we use entropy to describe the *state* of systems or whether we use it to study *changes within* **or** *differences between* certain systems through time and space as a consequence of flows and processes, i.e., *entropy production*.

The present paper examines how these different understandings and uses of entropy vary within physics and chemistry and adds up to impose a quite complex picture on biological systems. Here lies a great part of the crux of a semantic problem.

Meanwhile, the problem is not only a question of semantics. Introducing entropy as a concept in biology will affect communications even further, as many of the classical terms will change or even lose their definitions with respect to the various levels of the biological hierarchy under consideration. This is valid, especially when considering how the almost arbitrary choice of the ontological “elementary particles” used in “entropy” calculations varies and as a consequence influences any discussions about “entropy” and the respective reference levels.

We will not make any attempt to resolve all the issues in the area. In many ways, it is much too late for this since many confusing statements in the field have already been made. Usually, the reason for this confusion is an imprecise usage of terms and the associated statements about various phenomenologies arising from this usage. Therefore, we will try to indicate the sources of confusion in the field of the application of thermodynamics to biological systems in general. This is performed to raise the awareness of new readers in the area. As a more constructive contribution, we will conclude with some modest, initial proposals toward a clarifying adjustment of present terminology. This is seen as a necessary first step toward a possible resolution. We find it astonishing that this is necessary after more than 100 years of usage.

## 3. Sketching Entropy Controversies

As a starting point, two major issues seem to be forming the core of controversies in the area. At the core, we can identify, first, the important statements by Lotka in the papers mentioned above concerning the relationship between the size of structures, energy flows, and usage in biological systems, although the papers take a clear first-law stance and tend to ignore entropy formation and dissipation. From this an additional discussion arises, namely whether in nature there is a *final goal* or *telos* or whether there is at least an *orientor* indicating a *direction* toward an *extremum state* of certain system property acting as an attractor, such as the storage, utilization, or efficiency of any of the thermodynamic functions [30,31,32]. A second perspective to this point is that there also seems to be a major bifurcation between the ways in which thermodynamic concepts, such as entropy, are used. In physics and engineering, we find entropy being used both to describe the *states* of systems (alone) and also to study *entropy formation* together with process and path dependency. This deals with the necessity to involve such aspects as how changes in and between thermodynamic states take place, i.e., to involve process-oriented thermodynamics. Both directions have been taken and can be observed in the literature, but most often in situations where they have been applied separately. Already, at this time, we must make the remark that most probably, to fully comprehend biological and ecological systems, both approaches, “both sides of the coin”, are needed to give a truly holistic understanding of biological systems.
**(A)** **What do biological systems actually do?**

In ecology, the respective processes are investigated by succession theory, which tries to describe and understand the continuous unidirectional sequential change in the species composition of natural communities [33], which is often accompanied by typical changes in ecological state variables. Based upon the initial abiotic conditions, typical sequences of species can be observed, which, under specific external conditions, can strive toward a typical climax community [34]. Meanwhile, also at lower levels of hierarchy, we may observe similar developmental trends among the constituent components, namely the organisms’ continuous adaptation toward optimal exploitation of the resources offered by the surrounding environment.

Lotka’s papers [2,3] seem, somehow, to be at the crux of a debate on how biological systems (probably primarily organisms) distribute and utilize the available energy that is offered to them as outside–inwards gradients relative to the surrounding environment. Lotka [2] states:


*“In every instance considered, natural selection will so operate as to increase the total mass of the organic system, to increase the rate of circulation of matter through the system, and to increase the total energy flux through the system, so long as there is presented an un-utilized residue of matter and available energy”.*


We interpret this as meaning that organisms develop to build up more biomass, which in turn increases both the inflow and outflow of energies from the system; these energies include both chemically bound transfers as opposed to destroyed dissipated energies. The fundamental importance of solar radiation in driving the ecosystems receives only little attention. The increase in biomass and flows may occur as long as useful energies are abundant:


*“If sources are presented, capable of supplying available energy in excess of that actually being tapped by the entire system of living organisms, then an opportunity is furnished for suitably constituted organisms to enlarge total energy flux through the system”.*


This seems implicitly to mean that the development will be heavily influenced and eventually constrained by the amounts of energy supplied to it. Likewise, he makes important statements concerning selection and evolution:


*“This may be expressed by saying that natural selection tends to make the energy flux through the system a maximum, so far as compatible with the constraints to which the system is subject”.*


Meanwhile:


*“It is not lawful to infer immediately that evolution tends thus to make this energy flux a maximum. For in evolution two kinds of influences are at work: selecting influences and generating influences. The former selects, the latter furnish the material for selection”.*


This passage clearly demonstrates an awareness of the problems of extremum behavior and probably also a special arrangement of causes between levels of hierarchy. 

All in all, Lotka saw both the storages and processes and thus the changes to be of importance in being responsible for shaping the behavior, development, and evolution of biological systems. In an additional paper, he asserts that the outcome of evolutionary processes would be a consequence of the competition for free energy [2].
**(B)** **Do we talk about endpoints or just directions (orientors)?**

Immediately upon presenting such views, one is forced to ask the question: Where does the development and/or evolution of biological systems end, and what will be the best indicator to tell us when this final state has been reached? In classical or conventional thermodynamics, we actually find relevant statements only in the form of references to the imperfect conversion of energy ending up in *heat loss*, as first described by Carnot [35], which led to the formulation of the second law by Clausius [36], Boltzmann [37], and Gibbs [26], according to which an isolated system will develop toward an equilibrium state. Here, we have to be aware of the fact that some approaches to ecosystem theory [38,39] and criticize the conventional application of the physical equilibrium term in ecology because several approaches investigating ecological systems show that the exact opposite, the disequilibrium, not a balanced state, is a focal parameter of all living systems.

This conventionally comprehended equilibrium state in thermodynamics is represented in classical thermodynamics by an equiprobable distribution of its constituting components, atoms, or simple molecules. Meanwhile, some physicists see the two formulations—entropy as heat or entropy as a measure of distributions of microstates—as quite different from each other [29]. In isolated systems, this equilibrium, over time, will assume a state of maximum entropy, a term coined by Clausius in the mid-1860s, which later gave rise to the understanding of entropy as defining the direction of time’s arrow. In the physics literature, the time it takes for a given system to return to its equilibrium state is often referred to as *relaxation time*. Much of the earlier work was performed in order to understand steam engines [35,40], where pressure and volume differences were important factors in the process of getting work out of a system. The interpretation made on the basis of molecules that formed the core of thermodynamic interpretations of the dispersion of water/vapor particles, and pressure loss was based on the observation of the above-mentioned imperfect and irreversible conversion of energy. It is from these original considerations about the distribution of atoms and molecules that the problems emerge when transferring or reducing thermodynamic principles to other systems.

From a biological point of view, it is obvious that biological systems are far from being simple assemblages of atoms or small molecules and that they cannot be compared with steam engines either (although many mechanistic interpretations may be found in the literature). Biology is represented by systems that use other ontological units as their basis, rather than mere atoms or molecules, and they are constrained by other types of boundaries (other than adiabatic, etc.). Likewise, they are far from being in an equilibrium state where the actual situation can be represented by just assuming a random distribution of the ontological particles throughout the time-space phases of the systems.

Notwithstanding, much of the existing terminology has been established on the basis of simple isolated systems consisting of gas(es). We have thus inherited these terms as the basis for the many attempts natural scientists have made with the aim of formulating hypotheses and theories based on “thermodynamics” to gain a better understanding of the development and evolution of biological and ecological systems. In particular, the entropy concept has been found to be widely accepted as a framework for such discussions. At the same time, there has been a tendency to use the concept in quite diverse and undefined ways.

Meanwhile, the use of the term entropy has also become quite controversial, not only due to the many ways in which it can be used (on other ontological particles) but also because confusion arises from the many attempts made to establish a connection between thermodynamic entropy and the use of the same term in connection with various concepts of information, starting with Shannon [41]. This latter conflict will be excluded from the current discussion so as not to introduce any additional and unnecessary confusion. Meanwhile, it should be noted that many works in the area place themselves exactly on this border between the two theoretical directions, e.g., [42,43,44,45]. We will try to focus on what we consider to be within the thermodynamic domain only.
**(C)** **What types of systems dominate in biology?**

Above, we briefly touched upon the different types of boundaries used to describe various systems, and a short clarification is now necessary. We will here apply the commonly used terminology by giving a distinction between types of thermodynamic systems; this is agreed upon by most authors and textbooks:(1)*Isolated* systems, where no exchange of energy or matter to/from the surroundings can occur;(2)*Closed* systems, which may exchange energy (both receive from and give back to) with the surroundings but with no exchange of matter, i.e., open to energy flows only;(3)*Open* systems, which may exchange both energy and matter with the environment in which they are embedded.

From the discussion so far, it is clear that all systems in biology belong to the domain of open systems, which leads, as we shall see, to some distinctive and important consequences.
**(D)** **What are the consequences of types of systems on the boundaries**

While the above distinction between types of systems on the basis of the characteristics of their respective boundaries may seem quite trivial, the resulting consequences to the potential behavior of these systems are not. While isolated systems only possess one possible direction of development namely, that of evolving toward or returning to their equilibrium state (internal equiprobable distribution of particles), the other two types of systems can, and most likely will, take another direction. The time it takes for the system to come to (thermodynamic) equilibrium is, by definition, its “relaxation time”.

In the case of closed systems, where only energy flow from the outside is possible (but eventually may be supplied by different types of energy), it will be possible to use the inflow to structure the matter already enclosed in the system. If the structure receives a continuous energy input, it can, in principle, exist, as long as this input or gradient is maintained. But the system will not exhibit one important major feature—the capacity to grow. It may be organized in various ways according to internal and external constraints but will never increase in size.

In contrast, open systems, which are subject to gradients (flows of energy and matter in an inwards direction), will be able not only to form a structure that may grow both in terms of size but also in terms of the complexity of its organization, e.g., Ulanowicz’s distinction between “growth and development” in his first book on ascendency [46]. Thus, the most important feature of life seems to be its propensity [47] to increase in size or mass, expressed as a simple increase in biomass but, in addition to this, to evolve and develop new ways of investing these gains in new structures, new compartments, and new components that seem to possess a drive to exploit these gradients and their possibilities as efficiently as possible, e.g., Lotka’s papers.
**(E)** **Can systems be described in terms of both states and/or processes?**

We have already discussed the developmental trend of at least one indicator of state, namely the inevitable evolution toward increasing and finally maximum entropy in the distribution of particles of an isolated system. We have already commented on the aspect of the *relaxation time* as well, as this expresses the rate of entropy production during the development toward equilibrium and, therefore, it might be important in future discussions; for example, concerning the importance of this term in the dynamics of biological systems with a hypothetical propensity [47] to stay at or return to some (dynamic) equilibrium or balanced condition, which might apply at any or all of the various levels of biological hierarchy.

Meanwhile, for the other two types of systems, *closed* and *open*, we may be able to foretell only a little. Both may be seen as important pre-conditions for the emergence of life. Closed systems can establish structures as a result of imposed (energy) gradients alone. Most common examples come from physics, e.g., Benard cells, but also the structures arising as a result of chemical reactions, such as the Belousov–Zhabotinsky (BZ) reaction, which can be seen as belonging to this type of system. Meanwhile, for a system to establish the BZ reaction, we will need the presumption that it has been opened at a certain time prior to the reaction to allow for the provision of the right mixture of chemicals. The two systems are often mentioned in discussions in relation to dissipative structures, efficiency in the breakdown of gradients, etc. Both ways of establishing structures could have played a role in the emergence of life. The whole may be considered as emergent properties, the understanding of which was proposed by E.P. Odum [48] to be a research strategy to increase our understanding of ecosystems.

The real difference between these two cases of structures and structures established in open systems becomes evident, as it is realized that closed systems are constrained by the gradients imposed on them either from outside or inside. Eventually, they will die out when these gradients are removed and/or dissipated, e.g., relaxation time, and will certainly not be able to exhibit growth and storage.

So, the most important difference between the systems in this context will be that living, biological systems, which belong to the class of open systems, will be able not only to persist but also to grow, building up increasingly complex molecules that take on various roles during their lifetimes, as remarked also by Lotka in the papers quoted above. They also tend to be highly dynamic systems in which energies and matter are transformed and exchanged throughout time and space.

Having recognized this difference in the properties of boundaries, together with the associated consequences to the structures, the next question arising is: is there a better way to describe the situation in open, living systems? When using entropy as a concept, are we better off describing states only (using path-independent state variables from thermodynamics) or is a process-based approach better, i.e., do we need to use changes in the state variables (usually path-dependent)?
**(F)** **What special issues emerge that need attention?**

A combination of the issues raised above is responsible for the emergence of most of the major items of confusion in the area. To summarize, there are basically five issues that we consider to be the most important:(1)The interpretation of the *Boltzmann–Gibbs equation* and the shift in respective *ontological units* or “particles” while applying thermodynamics to different levels in the biological hierarchy;(2)The confusing concept of *negentropy*, indicating that entropy can be negative, which is impossible, although dissipative processes in biological systems can result in them developing into states of increasing improbability/decreasing probability;(3)The use of entropy to mean both *state* and/or a *flow/process variable,* as indicated in several places above and issue b) in this list;(4)The use of entropy forms (or free energy, availability, exergy) as *extremum principles.* Should the system move toward a maximum or minimum? And a maximum/minimum of what? And should we speak about *rates/acceleration* or *densities?*(5)The choice of *reference level(s):* when moving to levels of more complex systems, we may face a situation where situations “close to a true thermodynamic equilibrium” are not at all relevant any longer. Rather, we must define relevant “dis-equilibrium” conditions of an environment corresponding to the respective levels in the biological hierarchy;(6)The use of *entropy as information*, a misconception introduced by Shannon under the influence of von Neumann in the works on information theory (e.g., genome calculations) has also added up to confusion in the area.

However, we have chosen to exclude the discussion of the possible connection to information entropy because this is not considered relevant to the issues we wish to raise here. Meanwhile, the situation is understandable since when looking at the similarities between the equations (isomorphism) used in various calculations of entropy, information, and biodiversity in the sense of the Shannon, Wiener, or Weaver indices [49], it seems inevitable that the connection will not be made at some point. As a result, much confusion in the area has already emerged, and ignoring this perspective in the current discussion does not remove the already existing confusion. We will return to this point when dealing with hierarchical orientors.

In the following, we will concentrate on the use of the Boltzmann entropy as a central starting point, in principle, following the suggestions of Davison and Shiner [50] but carrying it slightly further by making use of the expanded version in the form of the Boltzmann–Gibbs equation, which is partially isomorphic to the biodiversity index, as mentioned above. Recently, it has been recognized that this form of entropy could be more accurately named “landscape entropy” [51], and we interpret this as an indication of an increased awareness of some of the controversies raised here.
*(a)* *The classic entropy equations*

Boltzmann [25], in a search for a Hamiltonian function (*H*) of thermodynamic systems, came to the conclusion that this was most likely to be proportional to the possible number of (micro)states of the system (*ω*) and that the relationship would probably also be some kind of logarithmic dependency indicated by an *l*, so that:(1)H∝lω

This later took the more familiar form (*W* replacing ω probably due to typography):(2)$$S_{B}=k_{B}\ln W$$
where *S_B_* now designates the Boltzmann entropy, *k_B_* is the Boltzmann constant, and *W* is the number of possible microstates of a given system, i.e., the number of possible configurations of “particles” that may be exhibited by that system. Most often, the natural logarithm is used. Which type of logarithm is used only matters when comparisons of systems are made and when conversions are performed easily by multiplication with a constant depending on the types to be converted. 

When reformulating the equation in terms of the probability of finding a certain state of the system, we obtain:(3)$$S_{B}=-k_{B}ln\frac{1}{W}=-k_{B}\ln p$$
where the symbols refer to the same conditions as Equation (2) above, and the probability of identifying one microstate 1/*W* is replaced with *p*.

Meanwhile, this equation is valid for a system consisting of particles indistinguishable from each other with the same probability. In this case, where we are considering different and distinguishable particles that do not share the same probability, we end up with the Gibbs version of entropy *S_G_*:(4)$$S_{G}=-k_{B}\sum_{i}^{}p_{i\;}lnp_{i\;}$$
where the index *i* indicates the types of distinguishable particles.

In basic thermodynamics, one normally works with a given number of particles in the form of atoms or molecules, which belong, for instance, to the set of physical elements. In real classic thermodynamics, this might even be delimited to ideal gases.

In applying thermodynamic analysis in the form of entropy to biological systems and ecosystems, we first need to recognize that the basic ontological units of these systems are far from those of ideal gases, i.e., molecules or atoms, as used in classical thermodynamics. In fact, many physicists will insist that we no longer talk about entropy as this term does not have any relevant definition under normal conditions of life. Below, while deriving other principles of phenomenological behavior for various levels of the biological hierarchy, we will, to start, simply refer to these ontological units as “particles” using the word in its widest possible sense, sensu lato.
*(b)* *Negentropy*

Another major source of confusion was introduced in the seminal paper of Schrödinger on “What is Life” [10], where he introduced the proposal that living organisms feed on *negentropy*. This formulation leaves, at least implicitly, the impression that it is possible for entropy to be negative. In spite of this term being an oxymoron, it continues to be used in literature, although what happens will be better understood as mutual constraints. The statement was believed to present a possible resolution to the observation that living organisms seem to be anything but randomly arranged systems. Rather, they exhibit an entropy state that is lower, i.e., less probable than that of a random distribution. Hence, “something” has to make it possible to lower the entropy state. The conclusion was obvious that this “something” was making it possible to change the distribution in the direction of a less randomized, less probable state, i.e., driving the living systems away from maximum entropy in a decreasing, negative direction; hence, this “driving factor” was (mis-)named negentropy.

What we are really talking about here is that for systems containing or receiving energy (e.g., paragraphs on closed vs. open systems), we can identify situations where the systems display a lower probability in their entropy distribution than equiprobability, i.e., maximum entropy. This means that:(5)$$S_{state}<S_{max}$$
where *S_state_* is the entropy state observed and *S_max_* is the entropy of a state representing a random, equiprobable distribution of particles.

One measure that expresses this difference was introduced by Evans [52], stating that:(6)$$Ex=T\left(S_{max}-S_{state}\right)$$
where Ex is the exergy of a system and *T* is the temperature. As *S_max_* is always higher than *S_state_*, the expression is always positive. The difference is sometimes referred to as *thermodynamic information* and due to its formulation, implies some possibly interesting links to information theory and also thermodynamics. As already stated, we will avoid discussions about this topic here. In the late 1970s, this equation was used as the starting point for the first applications of the exergy concept to ecosystems [53,54].
*(c)* *Entropy as states of statistical distribution(s) or dissipative processes*

Now, we may agree upon using the above expression (Equation (4)) as a measure of the distribution of particles in systems. Taking the classical case of an ideal gas under isolated conditions, i.e., surrounded by a boundary impenetrable to both energy and matter, the system will end up in a state where all particles are equally distributed. This is dictated by the laws of statistical dynamics, kinetic gas theory, and the models of random molecular dispersal demonstrated in basic physics. In such a system, the entropy *S_state_* will develop toward its maximum, *S_max_*, a state in which all particles are randomly dispersed. Thus, these conditions at the same time refer to a state of the system where thermodynamic information and exergy (work potential), according to Equation (6), equals 0 (zero).

But as said, neither biological systems in general nor ecosystems deal with atoms and molecules as fundamental particles and basal ontological units. Even in cases of extreme reductionism, we are unlikely to consider or accept this possibility. In other words, we have left the realm or domain of classical thermodynamics. The hierarchical levels of organisms must be accepted as realistic levels to be used as the basis for calculations utilizing the previous equations. We only need a shift in the ontological particles. This has happened several times in the history of the application of thermodynamics to biological systems. Meanwhile, the results emerging from these calculations cannot be considered to be representing classical entropy anymore.

In addition, we are now dealing with “entropy” states which, from time to time, may approach some stable, steady state, or homeostatic conditions, but in the intervening time, biological systems may often exhibit periods of high dynamicity. It is of course interesting to identify what influences (constrains) these dynamics, why and how the systems move from one state to another, and how the changes are reflected by the thermodynamic functions applied during these changes. Therefore, we also need to account for the path-dependent processes between states and what exactly their role is in bringing in the resulting changes. That is the two views that imply a major difference in units by introducing a time dependency for the processes. A thermodynamic description most likely needs to understand both states as well as the flows and processes involving the formation of entropy.
*(d)* *Maximum, minimum, or …?*

Two major directions of *thermodynamic extremum principles* can easily be identified in the scientific literature, namely a *maximum entropy* [55,56,57,58,59] or a principle of *minimum entropy* [23,60,61,62,63,64,65]. These two labels seem to imply a self-contradictory situation. How can entropy be maximized and minimized at the same time? Or are we dealing with a situation where shifts occur in the relative importance of these two principles in time and space? Are we discussing entropy as a state variable or as a process? And are both situations possible? Perhaps instead we need to talk about a compromise where different forms of entropies are *optimized*, i.e., balanced with each other aiming at increased efficiency regarding the prevalent and respective situations that a given system is placed in, for instance with respect to its life cycle. This situation is envisaged in several works in the field [66,67,68,69].

Thus, for various reasons, the discussion seems to have taken two fundamentally different directions: one (1) that follows the idea of maximization of an entropy-like function, which seems to be founded in the works of Ziegler and later Jaynes on inference problems [70], and another (2) based on the minimization of entropy for FFE systems, as formulated by Prigogine and co-workers [23,71]. Both directions seem, at present, to have reached a state where they are represented by various flavors that relate to the issues mentioned in the sections above, i.e., whether the expression under consideration is used to describe an entropy of state or the actual entropy production.

This leads to further questions within the paradigm of maximization of entropy. Many papers have pointed out that at least two or three basic understandings have been applied to these systems since they tend to refer to abbreviations or acronyms, such as first “MaxEnt” and “MEP”; both abbreviations seem to stand for *maximum entropy principle* and later a third term, “MEPP”, the *maximum entropy production principle*, was introduced. This is a rather simplified summary, as intermediate explanations are also found in the literature. The usage of these terms is far from being consistent between respective authors, and it is not always easy to interpret if it is real entropy, i.e., is it classical thermodynamic entropy that is referred to? Or is it actually a concept resembling entropy which is perhaps closer to information theory? Some remarks on the confusion that arises when implementing these principles can be found in the editorial by Kleidon et al. [57], as well as in Harte’s general introduction to the use in ecology [42]. Proceeding to the *minimization principles* presented in the literature, further confusion arises. As already indicated, one finds statements that seem to be exactly opposite to the ones just presented. Most prominent is the minimum entropy production principle (which unfortunately would come to share acronyms with one of the maximization principles). For this reason, we will refer to it as the Priogine–Wiame principle [23]. As stated, this principle refers to a rate, but unfortunately, it is not always clear whether the concept relates to an absolute, cumulative value or whether it should be normalized with respect to density, i.e., as a minimum of J/K per unit time and unit mass of the system under consideration. Regarding a corresponding principle stating a minimization of the entropy state of a system, we may identify Jørgensen and Mejer’s maximum exergy principle [53], which was based on Evans [52] expression (Equation (6)) for thermodynamic information:(7)$$\Xi =T_{k}\times I_{therm}$$
where Ξ is exergy, *T_k_* is temperature in Kelvin, and *I_therm_* is the thermodynamic information given according to the above:(8)$$I_{therm}=S_{ref}-S_{state}$$
where *S_ref_* is the probability distribution of particles at a reference state that needs to be defined and *S_state_* is the observed particle distribution of the system. This value can be a cumulative expression of the major constituent elements of the ecosystem, e.g., C, H, N, O, P, and S, which for instance, as stated by Morowitz [72], make up more than 99% of the elementary composition in biological systems. When the system moves toward a state of increasing exergy, it builds up more and more chemical elements. Thus, the deviation from maximum “entropy” increases accordingly, corresponding to a decrease in the “entropic” state of the particles.

A simplified set of acronyms is needed to illustrate the various situations described in the literature and investigate how they have been used. For this reason, we suggest that future contributions in the field should be as stringent as possible in implementing, for instance, the following suggested terms:*ME state* to describe situations where the principle used must be understood as ontological particles developing toward a situation of maximum heterogeneity and equiprobable distribution;*ME prod* will be used to describe situations where the system displays a maximum production entropy, an increase in exergy/energy destruction, or degradation;*JM principle*, referring to Jørgensen/Mejer, will be used where the system observed, together with its ontological particles, which will tend to deviate as much as possible from thermodynamic equilibrium; for instance, thermodynamic information and exergy or a properly chosen reference state (see section f below);*PW principle*, referring to Prigogine/Wiame—will refer to the principle of the minimum entropy production rate sensuo lato, not necessarily explicit time or density dependencies.

*(e)* 
*The choice of reference levels*


All biological systems exploit a gradient imposed on them from the external environment, to which they, in turn, must return all dissipations both in terms of energy (usually heat) and matter (usually small organic or inorganic molecules) [73]. The point here is that even this external environment deviates from the concept of a surrounding reservoir used in basic thermodynamics. In other words, the real thermodynamic equilibrium is not an operational reference state with respect to life conditions, i.e., the biosphere.

Therefore, we need to define an environment where it is still possible for life to exist. Most works in this area refer to some state of an inorganic solution exemplified by conditions in an “Oparin Ocean” [74] and have assumed, as a reference point, a hydrosphere where the first simple organic molecules have emerged, leading to a further increase in complexity and eventually to the first forms of life; this is a more rigid physicochemical approach that basically corresponds to a media where dissolved inorganic nutrients are found in the most oxidized states at (bio)geochemical equilibrium concentrations [54,75].
*(f)* *The use of entropy in analyzing data inference problems*

In general, the maximum entropy principle seems to have been applied mostly as a method based on information theory for investigating so-called inference problems in biology and ecology [76,77,78]. As previously suggested, we will omit the area of entropy as information from the discussions here and bring this statement only to indicate the connections that exist between the ME-based principles and the Shannon information/entropy concept, which seems to be tight in this area. One way of circumventing this dilemma is to accept various kinds of relationships between thermodynamics and entropy, for instance through the introduction of a distinction between structural and symbolic information, as suggested by Feistel and Ebeling [79]. However, this approach seems to physically disconnect the two entities, although it recognizes that there is a connection between information treatment and the flows of energy and entropy [80].
*(g)* *The use of thermodynamic orientors in ecosystem theory*

In ecosystem theory, several extremum principles have been observed. Based upon the above-mentioned approaches of succession, information theory, network theory, and applied thermodynamics with different viewpoints have been used to work out lists of ecosystem features that function as so-called orientors [30] or goal functions [81], which are optimized throughout undisturbed successional dynamics (see also [82,83]):


Community orientors:
-Biodiversity-Niche diversity-Life span-Body mass-Biomass-Symbiotic relations-Functional redundancy



Structural orientors:
-Information-Heterogeneity-Complexity-Connectedness-Gradient emergence and maintenance-Gradient degradation-Specialization



Thermodynamic orientors:
-Exergy capture-Exergy flows-Exergy storage-Total entropy production-Emergy-Power-Ascendency-Mutual information-“conditional” entropy



Ecophysiological orientors:
-Loss reduction-Nutrient retention-Storage capacity-Flux density and internal flows-Cycling-Respiration-Transpiration-Total system throughput



Network orientors:
-Indirect effects-Average trophic levels-Trophic chain length-Residence times-Network synergism


It is noteworthy that most of the descriptors used in the above = mentioned fields are directly or indirectly connected to equations with roots in “thermodynamics” sensu lato.

Of course, the production of entropy or the description of an entropic state should also be included among these orientor functions. While the state of ecosystems usually developed in an entropy-minimizing manner, we interpret this as the maintenance of disordered conditions—the succession in non-disturbed conditions increases the overall order of the system.

This long list of ecological orientors may demonstrate that the search for a correct valuation of entropy in ecology is based on fundamentals, disciplinary data, and knowledge, and the diversity of indicators may illustrate the demand to integrate developmental experience, e.g., within a basic thermodynamic theory.

## 4. Ontic Openness—An Intrinsic and Analogous “Entropy Driver” in Biology

Before starting our investigations on entropy, life, and biology relations proper, we need to explain one important feature of the observed phenomena in biology, namely a propensity that systems develop toward increasing diversity or complexity (or how one wishes to describe it). While in classical thermodynamics we have the kinetic gas laws, statistical mechanics, etc., to explain the evolution of a (gaseous) system consisting of atoms or simple molecules toward an increasingly randomized organization, we as biologists need an acceptable interpretation that explains to us that a corresponding phenomenological principle with a similar macroscopic effect for systems, which should come into play and be valid at all higher levels of the biological hierarchy. Considering the vast differences in ontology and particles between levels, we may need to search for several explanations.

But there is at least one fundamental property of biological systems that may at least serve as a partial explanation since this property leads to a general, intrinsic behavior that is quite similar to entropy and the distribution of particles in equilibrium thermodynamics. This is the property named *ontic openness,* introduced by W.M. Elsasser [84,85,86]. As the topic has been dealt with elsewhere [87,88,89,90], we shall only briefly describe its relevant aspects here. In short, Elsasser found a major difference that exists between physical systems and biological systems is that while the first was relatively simple and *homogeneous*, i.e., consisting of similar units), the latter was found to be highly *heterogeneous*, i.e., consisting of a variety of non-similar units comparable to the ontological particles used here. Thus, the systems must be regarded as much more complex. The resulting feature of *heterogeneity* has wide implications when searching for the phenomenological explanation requested above.

Most, if not all, biological systems are so complex that when applying *combinatorial calculations* based on constituent particles; for instance, to estimate the possible number of interrelationships at their respective levels of hierarchy, it is found that such calculations lead to so-called *numerical explosions* and quickly reach a level of possibilities that Elsasser referred to as “*immense numbers*”. Elsasser sets this level as being numbers greater than 10^100^ (googol). Such systems are intrinsically *indeterministic*. One consequence is that most realized states are *unique* and will never be repeated [82,91]. In principle, such systems are *unpredictable*, and this fact destroys all hope of finding deterministic solutions. Thus, this feature of biological systems represents a jump from a situation with normal fixed probabilities to Popper’s “world of propensities” [47].

As all biological systems display ontic openness, this also means that they behave in an unpredictable manner; each state must be considered unique as the chance or [47] probability of repeating a previous state is 0 (zero). This is valid at all levels of the biological hierarchy, from the genome through cells up to entire ecosystems, as any number of particles above 80 can be shown to satisfy the criteria for reaching immense numbers of possible interrelations [91]. The recent situation around the emergence of COVID-19 (an RNA string with a length around 30.000 with no proofreading mechanism) serves to illustrate the potential of such an intrinsically variable component in this type of system.

By introducing this feature into biological systems, we have also identified the variational component that is an important condition to evolution in neo-Darwinian theory. *Variation* is often just assumed to be present, as something for the selection mechanisms to work on, but it is often neglected when it comes to discussions on the deeper causal roots that give selection some material to work on. Ontic openness serves as an explanatory and causal component. It should be remembered that ontic openness includes no direction in the sense of improving or worsening the state of the system—either direction of the system’s evolution is equally possible.

It is remarkable that the probability calculations called for in the almost classical example used to explain entropy, irreversibility, and *Maxwell’s demon*, can easily reach an immense level and thus lead the system to be ontic open. In the case of an often-used classic explanation of entropy, we are presented with a box divided into two chambers. All particles are initially found in only one of the two halves. Now, a slit is introduced in the wall between the chambers, and the particles disperse, ending up in a situation of equiprobable distribution in the whole box. Dolev and Elitzur [92] calculate the probability that the distribution will return to the original situation to be in the order of 10^10^24^, which clearly indicates that such a system is ontic open. In the future, we may discuss whether the distinction between homogenous and heterogeneous systems is in fact a valid criterion that can be applied to detect the difference between physical and biological systems since the property of ontic openness seemingly also pervades into the area of physics.

## 5. Problems with Connecting to Far-From-Equilibrium (FFE) Conditions

The above-mentioned controversies (Section 2) are not restricted to the problems of a merely physical understanding of the systems. What we wish to do here is to investigate the process of extending the fundamental physical laws to also encompass natural systems. In this context, biological systems must be considered as open systems, existing under far-from-equilibrium (FFE) conditions. Most likely, it is not possible to understand such systems within the framework given by classical thermodynamics. It should be remembered that the FFE situations mentioned in many of the original works in this field are in fact far closer to equilibrium than the systems we intend to investigate here. Thus, many attempts to extend thermodynamics into the domains of biological systems assume that the systems represent some quasi-equilibrium conditions and display linear relationships between forces and fluxes of the systems [93]; hence, some of the additional problems are noted above. According to Ho et al., such close-to-equilibrium demands are fulfilled by higher-level biological systems [94].

In order to facilitate the discussion, let us assume that it is possible to work out a framework within which it is possible to accept that the entropy concept can be demonstrated to be valid when applied to FFE conditions and nonlinear relations between forces and fluxes.

Several additional conditions are mentioned in the literature that are required to be fulfilled within the process of expanding the theory. These include, for instance, the just-mentioned linearity [95] and also conditions of relative stability [96] and reproducibility [97], but with time, additional conditions will probably be found to be necessary.

Initially, the works of Prigogine and coworkers [23,62,98,99] were building on the shoulders of Lars Onsager [21,22]. Here, it was clearly stated that the minimum entropy principle was assumed valid in a region close to equilibrium, and it was also assumed that linear relationships existed between fluxes and forces. Here, we meet a major obstacle since modern biology and ecosystem theory see systems as being dominated by nonlinear relationships in the processes going on, in, or between the components of the system.

Reproducibility is a demand for the extension of the MEP principle [97], as the MEP principle seems to be applicable to such structures only. Lineweaver [97] states that such a principle applies to universal structures such as planets, stars, and galaxies, but the question remains as to whether it also applies to the evolution of biological systems.

All in all, many of the requirements stipulated by different theories and different authors will be difficult, if not impossible, to fulfill, both for theoretical and empirical reasons. Many additional aspects can also be mentioned, such as self-similarity, self-organization, autopoiesis, autocatalysis, and hypercycles, which are assumed to work as essential mechanisms behind the thermodynamics of systems. Many such properties may be viewed as examples of emergence—see Table 1.

In fact, most of the results of biological self-organization listed in the table can hardly be compared with the gas atoms or molecules of “classical” thermodynamics. The listed features are mainly outcomes of active relationships between parts, which again can be ascribed to the much greater internal complexity and activity. leading to the emergence of biological hierarchies.

## 6. Thermodynamics in Biological Hierarchies

We now seem to be ready to take a look at some of the many implementations of thermodynamics and the various uses of the entropy concept that can be observed in the current literature [100]. However, we will concentrate on illustrating some of the issues raised above rather than an exhaustive literature review.

The above concepts have. over time. been applied in various ways to many different levels of the biological hierarchy. Thus, applications have demonstrated the use of. For instance. the concept of entropy over a wide range of ontological particles, e.g., [101]. This is by no means strange since the application of the equations above to a particular focal level immediately implies that the ontological units at the level immediately below may be used for calculations; see Table 2 [102]. Meanwhile, this does not necessarily imply that the entropies we talk about are homologous.

Moving to the biological hierarchy, things are happening that force us to stress these issues. From traditional *hierarchy theory*, we know that we are changing scales in terms of space and time [103,104]. In addition, we may now see that at a certain state, we move from conditions where systems are embedded in each other to a state that is a composite (the ecosystem), consisting of components that, on one hand, belong to the set of organisms but, on the other hand, display differences in size and function that result in them working on quite a variety of scales of time and space, even though they exist within the same system. This gives an additional complexity to the ecosystem.

Nevertheless, as stated in earlier papers, a shift occurs in the structure of the biological hierarchy when moving up toward the level of ecosystems. We observe a change from systems physically embedded in each other to a situation where the hierarchy is constructed by researchers and is increasingly adapted to address epistemological issues (see Table 2). The table is constructed on the basis of the traditional view of the biological hierarchy as consisting of cells, tissues, organisms, populations, communities, and the ecosystem [102].

This fundamental shift takes place around the organism/population level. Up to the level of organisms, all lower levels are embedded in the upper levels and delimited with a physical boundary, e.g., membranes, connective tissues, skin, or exoskeleton. This embeddedness has the consequence that all organisms share almost the same basic functions, regulation of time dependencies, and inner biochemical relations. An attempt to illustrate this point is given in Figure 1.

Consequently, organisms belonging to the same species share, to a large extent, the same respective functionality and physiological time scales. This relationship is considered valid to the level of (meta-) population, but with communities or societies, the situation gets more complicated.

Moving even further up the hierarchy to the ecosystem, all living constituent components or “particles” can indeed be assigned to organisms (or rather composites hereof, such as populations). However, even for a simple system, these may vary considerably in their observed time constants in accordance with hierarchy theory [105]. Often, the entropy production rate of organisms is found to scale with the organism’s mass (M) to the power of ¾ [105], and combining this scaling with the frequently observed M^¼^ scaling of the organism’s physiological eigentime, PET (e.g., expressed by the number of heartbeats), results in a total entropy production per unit mass, which is constant over the PET and universal to all organisms. At the same time, this also represents a necessary condition for maximum efficiency, i.e., minimum overall entropy production. It must be remarked that the PET is not equal to the observed time scale. 

In the construction of an ecosystem in terms of thermodynamic relations, the difference in functional role between the respective species, e.g., being an autotroph, a heterotroph, or decomposer, seems to be of greater importance than the various “roles” with their respective functions. They must all act together as a whole (system) despite their different time scale values.

In “entropic” terms, the expansion to higher levels has two core consequences related to classical discussions in ecology about time and space scale relationships. Increasingly higher levels include more and more particles that most often are dispersed and belong to an increasing number of compartments. The increase in possibilities of spatial distribution serves to create an initial increase in “entropy”. After this, specialization comes into play and may allow for the importance of efficiencies to emerge, whereby evolutionary organization and optimization principles come into play. Thus, cells are not necessarily individual cells any longer but will, for instance, through epigenesis, develop in certain directions, taking on *habits* and becoming part of tissues or organs like the muscles, kidneys, liver, etc. It has been argued that such an organization maintains the lowest level of entropy [106]. Eventually, the organs in combination form the organisms at the ultimate highest level, including all the embodied systems below (Figure 1).

The jump to ecosystems consisting of organisms is defined as living systems in their environment [107]. We must, according to hierarchy theory, consider organisms, populations, or functional units as our basic ontological entities. Eventually, when moving up to even higher levels, we rarely consider single individuals any longer but view them as assemblies, e.g., populations that have (almost) similar properties. Therefore, time relationships also change in their relative importance. Where an organism, according to hierarchy theory, has a longer time scale compared to all lower levels, their inclusion determines a top-down control of the system. That is, they must all adapt to or subsist within a common overall time constant.

The situation is much more complex in the ecosystem, as its development, adaptation, and evolution are dependent on sets of interconnected functional groups, usually referred to as *trophic levels*, which all possess their own characteristic intrinsic time scales and, according to one hypothesis, a greater content of free energy; see Figure 2. The situation is rarely so simple.

A certain trophic level in a network may be occupied by organisms belonging to quite different species and thus vary in functionality and spatio-temporal scales (see Figure 3). All these different time scales must act together as a whole for the ecosystem to work properly. Together, they may serve as parts to co-determine the functional time scale of the ecosystem itself. Likewise, the role of the respective contents of available energies becomes less clear. In particular, the variable structure where upper levels are feeding on several lower levels, and the addition of a recycling element adds up to some unforeseen properties, e.g., Patten [108].

The organisms are now expanded into an ecosystem network with increased entanglement. Thus, the previous relationships are becoming more blurred. The fastest reactions are demonstrated by decomposers of dead organic materials, and this is common to all kinds of detritus, i.e., not depending on the origins from respective compartments of the ecosystem. At the same time, the time and energy dependencies become difficult to explain when the upper levels are interacting with several of the lower levels, e.g., using them as feed. At the same time, this type of diagram reveals nothing about any regulations, cybernetics, or semiotics, for instance as a result of competition or communicative processes.

The basic processes at all of the forthcoming scales seem to be rather similar. Additionally, due to external flows of exergy inputs and entropy outputs, we need an autocatalytic sequence of chemical reactions, which produce a certain meta-stability of the single compounds. Investing in these reactions will provide some self-organized “lifespan” for the attained structure and will produce some entropic decay, which is “leaked” and has to be absorbed by the environment.

As a system develops more and more complex reaction networks through time, it will become more statistically ordered. That process is accelerated by the creation of gradients and organized inclinations between locations of low and high concentrations along a certain spatial distance. These gradients are found to gain a certain meta-stability already in the non-living appearances of microspheres and coacervates (Oparin). Their functional reliability will increase with the assistance of the above-mentioned characteristics of living systems. The gradients are responsible for an efficient specialization by defining certain process spaces, organizing the flows between these units, and maintaining their structures even over multi-generational time horizons.

Typical biological expressions of gradients are the organelles at the cell level, epithelia, mucous membranes, or the skins of organisms. The transition areas of ecotones, boundary layers between ecosystem compartments, or the frontiers between the subsystems of landscapes are ecological gradients. Additional gradients are created by microclimatic distinctions and by the accumulation of chemical compositions, e.g., within phyto-mass or along the enormous concentration variations of the soils.

Each of these environmental gradients operates at a certain spatio-temporal scale. Hierarchy theory pronounces the combination of spatial extensions and temporal rhythms: broad-scale units with big spatial sizes operate at relatively low frequencies under steady-state conditions. For example, the modification of geological features needs a long time and produces relatively uniform patterns. On the other hand, small-scale units, like the microflora of a forest, are small in extent and display fast dynamics. The most interesting feature about these scale interrelations is that the small units have historically created the bigger process bundles, and as they have been produced, the broad-scale holons sensu Koestler [109,110] provide constraints for the initial small-scale processes, restricting their degrees of freedom, i.e., constraining the system for example through mutual information [91,111,112]. The two mentioned differ from other types of network complexity studies that are mainly qualitative, e.g., [113], by being quantitatively founded in either energetic or material flows.

Coming back to our gradient viewpoint, it can be stated that in all self-organized systems, an input of exergy, e.g., solar radiation in the case of our ecosystems, is used to build up a complex system of internal gradients [38,39]. The resulting concentration profiles can be understood as components of internal order: they are surmounting the initial, high entropic, and equal distribution patterns of the systems, thus increasing their exergy storage. Normally this creation of gradients is a long-lasting, long-term process. It is based upon several single reactions, steps, and activities, which are accompanied by the slow development of ecological successions.

At the same time, the maintenance of the stabilized interwoven autocatalytic cycles leads to an increasing energy demand, which can be detected by an increasing entropy production. This output process can be observed, e.g., by measuring CO_2_ emissions, the evapotranspiration of water, or nutrient loss by seepage and erosion. Those degrading processes can occur very rapidly and with spontaneously high magnitudes and amplitudes if the system is approaching instability. But high process rates of entropy production are also necessary to keep a very complex structure alive. Therefore, the discussed form of entropy production can be used both as an indicator for the conservation of complexity and as an indicator of functional vulnerabilities.

Another interesting aspect of these eco-thermodynamic theories arises from the fact that the ecological target state is not the equilibrium, as required by all the basic physical comprehensions of entropy, but is at a disequilibrium. The more complex the gradients of a developing system are, the smaller will be the uniformity and similarity of the subsystems. Ecological succession, therefore, seems to reduce the embodied entropy of ecosystems, while entropy export is supported, as well as exergy storage.

In the following sections, we will attempt to deal with specific aspects of various levels found in current literature.
**(a)** **Cellular level**

Biological cells represent the level of integration where we are closest to understanding the system on the basis of thermodynamics alone, for instance by applying a strictly thermo-chemical viewpoint. Several attempts can be identified where cells have been viewed as factories or machines with a network of processes consisting of possible conversions between both simple and more complex organic molecules. The processes are often separated functionally in the organelles of the cell. The conversions are taking place in accordance with free energy and supplies of ATP. This information may be enough for understanding the basic functions of the cell, but it is not enough to understand life. This problem may be illustrated by papers where thermodynamics have been coupled to explanations of the emergence and evolution of life [18,114,115,116]. Cells are close to the simplest biological systems we have today, but some protobiological subsystems have probably existed. Still, cells are FFE systems, and this already triggers considerations and questions as to whether the classic concept of entropy applies under such “simple” conditions.

Within the cells, processes take place in organelles, which make up the sub-level components of the cells. Their very existence makes it possible for further ordering the processes so that specific processes that may not coexist can take place in separate organelles. The reasons for this spatial separation range from mere physicochemical considerations to a necessary separation of processes that would have too high a risk of interfering with each other or are not able to take place within the same compartment, e.g., in some processes, oxygen is necessary, while it is toxic to others. This compartmentalization of subsystems initially leads to an increase in “entropy” but may later serve to open possibilities for a further separation of processes. In the end, events involving compartmentalization tend to further increase complexity at the cellular level. At the same time, it seems that this tendency is accompanied by a decrease in entropy formation [117,118].

Eventually, a cell is not just a cell and, even at this simple level, we find a few basic differences that are important to our understanding of the processes going on at higher levels in the biological hierarchy. We must point out here the role of autotrophic components. In the widest sense, organisms are, in principle, able to live on their own, either using light as a primary energy source (photoautotrophs) or obtaining primary energy from materials and chemical compounds (chemoautotrophs). In both cases, oxidants are needed to synthesize the organic molecules necessary to ensure life and the existence of the cell or (simple) organism. Organic molecules are easily described by the *free energy* content (usually ΔG). The more complex the molecule, the higher its free energy, and the more the molecule deviates from equilibrium or another reference state. This is also reflected in the determination of *Gibb’s free Energy* from the calculation of equilibrium constants in thermochemistry.

Due to the restricted space, the thermodynamic relationships in the cells are difficult to measure. Most studies have been carried out using models of biochemical cycling of the systems in order to understand the macroscopic functions. Meanwhile, there is a risk that such models, for instance the one presented by Demirel [119], will be too reductionist to give us a simple overview in terms of a holistic thermodynamic balance.
**(b)** **Collection of cells, tissues, and organs “sensu lato”**

Although some mono-cellular organisms (e.g., phytoplankton) are abundant (sometimes as colonies), life soon developed into multicellular organisms both in the autotrophic and heterotrophic branches of the evolutionary tree. Some major differences can be observed if we consider mono- versus multicellular organisms. The difference is most likely to be found in the way the necessary material inputs of nutrients and respiratory elements to be used as oxidizers are supplied to the respective cells, i.e., in the relations with the external environment. In the first case, matter, often in the form of simpler molecules, is supplied through the relatively simple processes of diffusion or phagocytosis. The principles behind the organization of internal functions are surprisingly similar. Early in the development of multicellular organisms, cells became differentiated to carry out different higher-level functions, forming tissues and developing further into digestive tracts, excretory organs, transporting “organs”, etc. These “organs” have been developed and refined during evolution throughout the animal and plant kingdom.

The thermodynamic effect of this specialization has only been analyzed in very few cases, and research studies using modeling seem inconclusive in this context [120]. In more complex organisms, the organs appear as highly structured and specialized tissues with individual functional contributions to the life of the organism, which at first glance may appear to be costly in a thermodynamic sense. However, it is only meaningful to evaluate their true value at the next level of complexity, namely the organism.
**(c)** **Organism level**

Organisms represent the ultimate level of nested systems, i.e., where all lower levels are somehow included or physically embedded in the *focal level*. All basal units are considered to belong to a class or set of organisms. One focal level n has the elements from level n-1 as ontological units. (Figure 1). Thus, at this point, the physically embedded type of scalar hierarchy ends, and we as observers need to construct the hierarchical levels and their respective ontological components, which we consider making up the system in accordance with other principles that we need to define ourselves.

The function of the individual organisms, also in the thermodynamic sense, is for all types of organisms determined by the dynamics of the inner components, the organs. Whatever the type of organism (e.g., autotroph vs. heterotroph, homeotherm vs. heterotherm), these parts are stated by many authors to interact and eventually reach a balance between the organism and its environment, which is often referred to as homeostasis [121]. Most likely, this overall state is established as a consequence of an average time scale, including the consequences of limiting functions similar to *Liebig’s law*, which hypothetically implies that the slowest reactions of the systems of the hierarchy below the focal level will determine the overall rate of the organism.

Meanwhile, in a thermodynamic sense, the entropy arising from all metabolic processes ends up as heat and/or excretory products. Both may be seen as dissipations, either of energy or matter, and need to be exported from the organism, passing over its boundaries and eventually ending up in the environment.

Taking into consideration the energetic balances of organisms as presented in (eco-)physiological textbooks and combining this with Odum and Pinkerton’s considerations on the maximization of power [4], the picture becomes increasingly complicated. When calculating thermodynamic balances, we really need to know which state of a given organism’s life we are working on.

One generally observed trend goes in parallel with productivity: organisms tend to increase their entropy production during epigenesis at a relatively high rate while they are young. Subsequently, the rate of entropy formation decreases [69,122]. The same seems to be the case when considering the relative complexity of organisms in the biological hierarchy, as the specific entropy is found to decrease with the level in the hierarchy from yeast to birds [123].

While the maximum entropy principles are often argued to be valid also for organisms, the conclusions are often drawn from the fact that larger organisms have a higher dissipation. This is in accordance with Lotka’s descriptions, where larger and more complex structures exhibit a higher throughput. Meanwhile, the previous authors seem to confirm that even though the dissipation is larger, it happens at a lower cost per unit, in short with an increasing efficiency. The findings of Vallino and Huber [124] that the microbial populations in a meromictic pond develop according to the principle of maximum entropy is equivalent to the results of Ulanowicz [125,126], where the recycling component is allowed to have higher dissipation than the average of the component organisms and yet contributes to an increase in the *ascendency* of the system.

The relatively new discipline of dynamic energy budget modeling for organisms may open new possibilities for thermodynamic analysis at the organism level; this will be interesting to follow in the future [127]. The same is valid if we consider applying recent findings from the relatively new discipline of bio-semiotics [128] or even ecosystem semiotics [129]; this consists of including the role of communication, in its widest sense, among constituent organisms and the role of this communication in shaping the thermodynamic properties of the upper-level systems. For instance, investment in pheromone emissions has been found to facilitate foraging, “maximizing energy gains”, and increasing efficiency toward stabilized conditions [130].
**(d)** **Collections of organisms—populations and communities**

The next levels of the biological hierarchy met in ecology are the population and community levels. Populations are collections of organisms usually belonging to the same species, whereas a community refers to a functional grouping, e.g., a microbial community, plankton community, plant community, and so forth. Both serve to illustrate the consequence of taking thermodynamics to a level where it is necessary for the observer to construct the boundaries of the system.

The boundary of a *population* may be quite virtual but is often defined in bio-geographical terms. This means that a boundary depends on an area that is normally exploited by the organisms. No need to say that this is much easier to define for plants than animals. In either case, the dynamics are determined by the intrinsic characteristics of the organisms, their life cycles, and physiological time scales.

It follows that the concept of *communities* is much more complex and may not be very useful for a thermodynamic approach. First, the fact that the constituent organisms at this “level” do not necessarily belong to the same species introduces multiple possible variations in their respective time constants. Second, they do not necessarily share exactly the same functions and hence may act differently with respect to the exploitation of both matter and energy gradients. Taking the example of microbial communities often mentioned in the literature, for a given soil or sediment, we will experience a set of quite different oxidizers and corresponding (inorganic) nutrients, as well as dominant processes with corresponding differences in energy consumption and dissipations.

Much of this complexity is transferred to the next level of hierarchy—the ecosystem. The question is whether we can assume a general trend from cells to ecosystems at the slowest physiological time at lower levels of hierarchy, which dictates recycling and hence turnover/development at the upper levels (and the ecological time scale-related version of the Liebig laws)

What happens in a thermodynamic sense when organisms of a certain species gather into populations or communities? Does this new structure simply act as an additive feature so that entropy production can simply be summed over the number of organisms, or does it present a relative increase or decrease in entropy production due to additional emergent properties?
**(e)** **Ecosystem level**

The ecosystem level differs since by definition it is a composite of organisms, populations, and abiotic entities, which may all vary considerably with respect to their time constants and space scales. The variation exhibited is now likely to be even larger than in the above-mentioned example of communities. In addition to simple bio-geochemical functions, we have to add the perspective of the ecosystem’s food chain or network [131]. According to van Dyne [102], “the functions of an ecosystem include transformation, circulation and accumulation of matter and flow of energy through the medium of living organisms and their activities and through natural physical processes”. All in all, it seems feasible to apply a thermodynamic viewpoint to interpret the behavior of ecosystems. One question left is whether we really need to discussions if the ecosystem is a “superorganism” or if it is just a sum of individual behaviors [132,133]. This issue has often been dealt with under discussions about the emergent properties of ecosystems. Such properties appear already at lower levels, and their importance only seems to increase with complexity.

The most common way of constructing a hierarchical view is to concentrate on the pattern given by “who eats who”—usually referred to as the *trophic chain*. This all starts at the level of the primary producers and continues through herbivores, primary and secondary carnivores, etc., forming a functional, scalar hierarchy where each upper level feeds on the level immediately below. In some rare cases, the ecosystems are so simple that they are composed of a single trophic chain, but usually, they consist of more or less entangled networks, as already pointed out by Lindemann [131]. One focal level feeds may even feed on several of the levels below. In addition, depending on the respective time scales of the organisms as they die off, they will enter one or several pools of dead organic matter (detritus) and, in association with these pools, we will find decomposers (bacteria) whose function is to break down the dead matter and thus ensure the circularity/closedness of the system.

In all cases, in thermodynamic terms, the ecosystem must be considered to be an open system, as it needs energy either in the form of (1) solar radiation or (2) bound in matter, or most often a combination of the two, to exist. In both cases, they are constrained by this energy; in the first case by energy captured by photoautotrophs through photosynthesis and in the second by external supplies of allochthonous material or the internal recycling of nutrients made a possible detrital component.

Ecosystems have been analyzed from almost all types of second-law thermodynamic perspectives including entropy [134,135,136], maximum entropy and maximum entropy production [137,138], minimum entropy [139] and exergy [53,54], and eco-exergy [81]. The latter perspective includes the suggestion of integration with a complexity measure of the ecosystem components and thus represents a possible link to information and energy-based indicators.

While in basic physics, gas molecules are used as basic units, we need other basic particles when we move out of the organism scale. Of course, it is easy to say that we need to account for all the organisms as units. When doing so, we have to be aware (i) that viruses and bacteria belong in this class, and we do not have sufficient knowledge about their appearance, activity, and structure of their community. In addition to this point, real ecosystem comprehension must also account for (ii) the abiotic processes, the flows, storages, and effects of energy, water, and matter transitions. If all of these components are considered in a holistic manner, we will find an enormous complexity of elements, subsystems, and scales, which hinders us from defining the “particles” or discovering an easily understandable and measurable common dimension. Furthermore, we can arrange the collection of basic units on the basis of functional features, e.g., referring to the focal pools and flows of water, energy, carbon, and nutrients. In that respect, the particles would also include soil horizons, groundwater storages, or specific microbial biomasses and soil organic matter (SOM). A holistic approach must integrate all these functional aspects, as well as the structural subsystems discussed above.

## 7. Emerging Changes through Hierarchies—Entropy in a Macroscope

So far, we have, in principle, considered the thermodynamics of organisms alone, so that the thermodynamics of populations is represented by the sum of thermodynamic relationships in a given environment of a set of organisms belonging to the same species. In the same way, the ecosystem is composed of populations, and its thermodynamics potentially is merely the result of the thermodynamic relations of all species. So, it is now necessary to ask the following question: it is really all that simple?

According to the many writings from the classical “Fundamentals of Ecology” textbook by E.P.Odum, [5] via the sketching of ideas like the eMergy of H.T. Odum [140], the (eco-)exergy of S.E. Jørgensen et al. [81], and through today’s efforts in the area of ecological indicators, most systems ecologists and ecosystem theorists will tend to state that such a standpoint is too simplistic. “The whole is more than the sum of the parts” and “everything is linked to everything” are common statements used among ecologists, clearly demonstrating the attitude that the situation is much more complex. Ecosystems exhibit emergent behavior.

All the areas mentioned above contain some sort of thermodynamic elements that need to be analyzed. Some of the core target points have been mentioned already, and it appears that we need to refine our language in order to communicate properly around this. At the core, we find both organisms and the ecosystem, which seemingly tend to become larger and larger, accumulating free and available energy and spending it in more and more complex manners with age and time throughout their development. The larger the structure, the greater the dissipation(s). Both sides of this observation seem to be covered by both the eMergy and exergy theories and very much relate to Lotka’s original formulations on this topic. The question arises again: is it energy, exergy, or entropy relations maximized, or is the outcome something in between? In order to investigate this, we need more insight into what is actually taking place with the energy fluxes in the systems.

At the ecosystem level, some relatively new ways of investigation have been appearing through the application of network theory by B.C. Patten and R.E. Ulanowicz and their respective coworkers. First of all, Patten has demonstrated both the quantitative and qualitative importance of flows in the network, and that a major part of the energy in the network is made up of the internal flows [141]. The “cycling” of the materially bound energy leads to intricate patterns and rules and exhibits some surprising (emergent) properties. Ulanowicz has been working from a similar starting point and investigated how the ecosystem network is likely to behave over time [46,91]. This has led to the formulation of a property called *ascendency*, a homolog of Gibbs free energy, where the distribution of flows is expressed as the average mutual information of the system. In short, the flows and how they are distributed is important. Systems will tend not to be too overly complex or too simple. Probably, in the first case, because too much energy becomes wasted because of inefficient organisms, and in the second because the system becomes too fragile, as it contains no “backups”, they can meet the challenges in an ever-changing environment. 

Both network theories contain important messages about the thermodynamics of ecosystems and merit more attention on, for instance, the direction of analyzing the evolution in the efficiency of flows as compared to the sizes of the systems. At the same time, the storage perspective should be addressed and compared to eMergy and the early versions of the exergy contained in and destroyed in the system at various locations in time and space. According to Sciubba [8], the energy in living systems “is never considered as per its *content* but as per its *flowing*”, which confirms the tendency for storage to be overlooked in this research.

## 8. Why New Additional Entropy Interpretations Are Needed!

There is no doubt that the application of thermodynamic viewpoints in biology and ecology in general has met with a rather reluctant acceptance in several areas of the natural sciences. Physicists are too holistic since they claim that is it not possible to extend physical laws developed for natural gases and molecular kinetic theory under conditions close to equilibrium to apply to conditions far-from-equilibrium since, for instance, entropy is not defined under such conditions. So, how may we justify even using the term entropy to the level of complex, conglomerate systems that are clearly so far-from-equilibrium conditions? One is tempted to say as far away as possible from equilibrium.

From many ecologists, the response would be that such comments are much too reductionist, as laws developed on the basis of molecular systems will never be able to cover such complexity as represented by, for example, organisms and hierarchical levels above them. Again, others will argue such an approach is much too holistic, as the functional actions are dictated and determined very precisely from the lower hierarchical levels, e.g., the genes. Such a viewpoint somewhat represents an extreme version of reductionism. Skene [142] argues that the reason for this is that the “academic foundations differ significantly, with the modern evolutionary synthesis” at one end, whereas “ecology more recently has utilized a system theory approach”.

Meanwhile, experience from our careers as scientists and practical empirical researchers within ecosystem studies has taught us that a thermodynamic approach, even if applied in an intuitively based manner, can demonstrate some important principles that would otherwise be forgotten or ignored. In addition, we agree that approaches are needed to build “a bridge between ecological studies and evolutionary biology”, which necessarily can serve to integrate human society and our management of the biosphere [142].

One important issue is that the basic acquirement of energy in the systems always occurs from the outside to the inside, i.e., in an inward direction. This emphasizes the importance of the *input environ* sensu Patten [110] as an ultimate driver of the development and evolution of biological systems. How the system eventually evolves or develops as a result of internal relationships, which also include the ultimate recycling of energy in matter via detrital links, a mixture of dead organic material, and microbes. This link is often regarded as the ultimate lowest level in a thermodynamic hierarchy of the ecosystem’s food chain or network. Nevertheless, this link may be considerably more important than expected [125]. The findings stress the importance of making a distinction between the *output environ* according to Patten, splitting the transferred energies into two parts, one that is useful within the system and to adjacent systems and another that is no longer useful, i.e., dissipated. By working with exergy, the picture is likely to become even clearer.

As the appropriate inputs and outputs are brought into focus, so are the size of the environment and the boundaries. That is, the surface-to-volume ratios, or in the two-dimensional case, the circumference to (internal) area, need to be considered in order to understand the thermodynamic balances of the systems. The exchange with surroundings is facilitated by high surface-to-volume ratios, which often result in a relatively higher intrinsic capacity for growth within the systems. This is a common rule of thumb that seems to be independent of the level of hierarchy. On a larger scale, however, this also makes the ecosystems more vulnerable to constraints imposed from the outside. For instance, small, isolated patches are much more easily affected by invasive species.

An additional observation is the historical development of the ecological sciences, which in its earlier stages was founded very much on the observation and registration of the constituting particles of the ecosystems and was related to states rather than processes. The philosophical aspects were addressed earlier by Whitehead, who stressed that systems are not equal to their components but are rather what these components do when working together. The importance of processes is accentuated not only in thermodynamic studies but also in the application of network theory to ecosystems in the works of Patten [110,141,143] and Ulanowicz [46,91]. One major difference between the two approaches is that system drivers are seen as external versus external constraints, respectively.

In many papers, it has been suggested that ecosystems either maximize or minimize certain functions or properties, and this necessarily leads to a logical question: do we really mean maximum or minimum as a final state (indicator) of the system, or do we only mean that the system possesses a tendency (propensity) or “desire” to develop in a certain direction (orientor) toward a “telos” that it will probably never achieve [144]? Some recent findings may suggest that ecosystems somehow, through their function within a real world full of interrelations and disturbances, attempt to stay on the safe side of chaos or breakdown. The price paid for this is that they sub-optimize and never reach any ultimate maximum or minimum since this will necessarily bring the system into areas of instability [145]. Instead, they tend to stay at a level of robustness, as indicated by Ulanowicz [91].

## 9. Recommendations for Future Works—Preconclusive Remarks

From all the above, it is clear that there is a need for greater precision in the language we use when discussing applications of thermodynamics to biological and ecological systems. These improvements should particularly focus on defining what we actually mean when we use a word like entropy. The specifications should aim at creating a healthier and more fruitful platform for discussions on this topic. In future reports of studies in which we use thermodynamic approaches to analyze the states, behavior, and evolution of ecosystems, we researchers should maintain a high level of stringency and consistency when discussing the measurements and calculations we have made.

*Firstly,* it should be clear what “type of entropy” we are expressing, and secondly what “elementary particles” we use as a basis for our calculations. One immediate and obvious suggestion, arising from our readings in connection with the development of this paper, would be to relate the “entropy” concept that is used to the focal level of analysis; for instance, we must specify:(a)Biomolecular State Entropy vs. Biomolecular Entropy Production(b)Organism State Entropy vs. Organism Entropy Production(c)Population State Entropy vs. Population Entropy Production(d)Community State Entropy vs. Community Entropy Production(e)Ecosystem State Entropy vs. Ecosystem Entropy Production(f)Landscape State Entropy vs. Landscape Entropy Production(g)Biosphere State Entropy vs. Biosphere Entropy production

It is probable that this listing is insufficient, and that additional “entropy” and varieties of “entropy” will appear with the increasing use of the concept in new fields or areas not covered in this study. Examples of the range of implementations may be found in Jaffe and Febres [146].

If such a common terminology can be established, it would greatly assist future discussions, in particular when talking about states, absolute rates, or rates that have been normalized; for instance, with respect to size and, debates about the use of any of these “entropies” as extremum principles.

*Secondly***,** therefore, we need to be clear about *reference levels* and definitions of equilibrium. We need to agree on common ground for comparison. This is relevant at all levels when working with “entropies” or other functions derived from thermodynamics such as exergy, eco-exergy, etc. For the evaluation of biological and ecological systems, a properly and precisely defined reference level will probably suffice [147].

*Thirdly*, when applying these concepts, we need to be clear about whether we are working with productions, rates, contents, and/or density-based expressions, and whether dissipations refer only to thermal entropy or also exports of smaller molecular compounds, e.g., X-entropy [148], partial destruction of energies, or exergy degradation.

It is our hope that the above problematization of the current literature will create an awareness of the problems that arise when studying it, and in particular, when we attempt to compare the various analytical approaches. This is very important for researchers working with the existing literature and newcomers in the area in particular. Meanwhile, observing a certain level of stringency in the future will surely help to clarify and hopefully resolve some of the issues that are currently under discussion in connection with the role of thermodynamics in living systems and all aspects hereof.

## 10. Suggestions for Connections and Possible Initial Steps toward Resolution

When evaluating the above attempts to analyze the wide range of applications of the concept of entropy, it must be concluded that the term does not have a uniform, unambiguous usage. In general, most of the problems can be reduced to questions of a usage describing states versus processes combined with situations of stability and linear–nonlinear dependencies between forces and flows. All of this leads to problems when analyzing biological systems from a thermodynamic angle. In the following, we assume that it will be possible to resolve questions about how far-from-equilibrium we can consider entropy to be valid and examine linear vs. nonlinear dependency.

Firstly, many applications refer to physical systems, which may be treated as (quasi-) isolated or closed systems. As a result, the treatments often involve exchanges of energy with the environment and do not involve exchanges of material. This is well-illustrated by the classical equation from Prigogine that is often used by scientists to describe the entropy balance of systems both from the direction of maximum or minimum entropy production, i.e.,
$$d_{tot}S=d_{i}S+d_{e}S$$
where *d_i_S* and *d_e_S* describe the internally formed entropy and entropy exchanges with the surroundings, respectively, and *d_tot_S* is the universal change in entropy, which must be positive. The problems in describing a negative change in entropy with the environment have been discussed above. All in all, this equation ignores the results of the processes, which may be valid in isolated and closed systems where the emergence of dissipative structures only exists as the result of energetic gradients imposed on the system, i.e., the structure disappears soon after the disappearance or exhaustion of the gradient. It is clear that the equation is insufficient when it comes to describing an open system where it is possible to build up a material structure that lasts for a longer time.

Secondly, therefore, we need a description that accepts that the system is in fact an open one, receiving energy in the form of matter (chemical energy). At some places in the discussions by Lotka [2,3] on the energetic relations between structures, it is possible to identify a partial identification and possible solution to the problem, in which a unifying principle must include at least three elements: (a) the driving power, i.e., input of energy, (b) the build-up of structure (measured as energy deviation from a reference state), and (c) the dissipations by metabolization of energy (exergy destruction) or export to other compartments of the system (see Figure 4).

At the same time, it would be desirable to couple this description to some basic (first-law) energy balances as normally used in physiology but reformulated in terms of exergies, e.g.:$${Ex}_{imported}={Ex}_{stored}+{Ex}_{respired}+{Ex}_{excreted}+{Ex}_{defecated}$$
where *Ex_imported_* covers import of radiative fluxes by plants and ingestion in animals and the digestion of matter by bacteria, *Ex_stored_* is represented by both the growth of tissues and reproductive investments, *Ex_respired_* is energy loss (dissipation) through any metabolic pathway (anabolism, catabolism), *Ex_excreted_* is loss through matter (exchange of solutes), and *Ex_defecated_* is the loss of the undigested part of ingestion (only animals). In this way, the traditional formulation has been adapted to cover both plants and animals, as well as simpler organisms.

When compared to Lotka’s model, *Ex_imported_* is comparable to the maximum power gradient available to the system, and *Ex_store_*_d_, over time, represents the build-up of a physical structure (biomass), including reproductive investments. The remaining elements represent dissipative pathways, where *Ex_respired_* can be considered to represent the costs of building up and/or maintaining the structure.

Now, it becomes important how we estimate the growth in structure. One possible way goes through Evans’ [52] equation (Equation (6)). From this, we could estimate a build-up of biomass (structure) over time from t1 to t2 by replacing the *S_state_* in the Equation with the entropy states at either of the two times, and refer to them as *S_state_*_,1_ and *S_state_*_,2_, respectively. In this situation, the change in structure could be estimated by a change in Exergy, *Ex*_1,2_ as:$${Ex}_{1,2}=T\left(S_{ref}-S_{state,2}\right)-T\left(S_{ref}-S_{state,1}\right)=T\left(-S_{state,2}+S_{state,1}\right)$$
where *S_ref_* is described by an appropriately chosen reference situation and the system is in thermal balance with the surroundings. It is assumed that the state of the system is continuously evolving toward less entropy, and thus the difference in the last bracket is positive. When seasonal variations are considered, there is a high possibility for ecosystems that neither of the latter two assumptions hold.

The equation could be translated into differences in free energies (availability) following the layout given by Lotka. When combining the two equations, it becomes clear that it is possible for a system’s state entropy to decrease while the sum of the dissipative losses derived from the exergy equation still ensures that the second law holds.

It should be noted that the above equations neither mention nor assume the minimization or maximization of any relationships, but rather deal with the evolutionary trends (propensities) of a system, so they describe the directional tendency of the system, which is given by its intrinsic possibilities (e.g., adaptation) together with the prevailing situations of the system’s environment. Eventually, it has been suggested that the two directions are “different viewpoints of the same aspect: the first (minimization, authors’ comment) is related to the system, while the second (maximization, authors’ comment) is related to the interaction between the system and the environment” [149]. However, this statement does not necessarily imply an optimal use of resources.

In fact, several authors mention the issue of the application of extremum or optimum principles in biological or ecological systems. In most cases, they point to the constraints of the systems, both internal as well as external, which, when taken together, seemingly prevent the systems from reaching such extreme states. Rather, the state achieved will represent some sort of compromise. So, the optimal path, as indicated in Figure 4, will be followed as closely as possible in accordance with both internal and external constraints. This is probably an internal defensive mechanism developed to counteract the typical properties of FFE systems. According to Prigogine [14], “Far from equilibrium, systems enter into the nonlinear range and display a multiplicity of solutions to the equations describing their evolution”. He continues his remarks by stating that the systems become sensitive to fluctuations and will bifurcate to states, which may not be predicted from deterministic equations [14].

Nevertheless, the vast majority of the limits to development belong either to the input of available energy or the way in which energy transformation takes place and how efficiently it is used. The actual evolution and development of the system is the result of the dialectic interactions between the two. We should, therefore, rather talk about a Balanced Exergy Entropy Principle (BEEP), where systems tend to avoid going over the ”edge of chaos”. Continuous refinement and fine-tuning of the function in ecosystems are known potentially to lead to crashes (e.g., ascendency studies) or breakdown as a consequence of the system’s lack of flexibility and passivity due to the high costs of complexity, which leads to hypothetically repetitive Holling cycling [150]. Such destructive dynamics would rapidly increase entropy production and reduce exergy storage, a situation that is not desirable. Therefore, ecologists can be happy that we seem to observe far fewer discontinuities than we might expect.

Coming back to our initial questions, we can finally summarize the discussions within the following points:(1)The use of the entropy concept, in general, has been continuously introduced in a wide range of areas of natural sciences, and much confusion has been introduced with these multiple adoptions. The reason for this disorientation is the produced mix of viewpoints of the disciplines from which different aspects of entropies have been derived, e.g., mathematics, statistics, physics, engineering and biology, or ecology. An additional reason for the “entropy in comprehending entropy” arises from the different types of systems where different varieties of the concept have been used. That diversity of systems is highly correlated with a diversity of scales, and we have seen that the outcome of entropy analyses can be extremely different if different levels of the analytical hierarchy are used.(2)One major distinction between the various uses can be derived from the basic starting points of the analyses. On one hand, we find entropy comprehensions, which, in their basis, are related to the distribution of ontological particles of the system and thus potentially describe the state properties. On the other hand, we find entropy concepts that are more closely related to material and energetic flows and conversions, and thus are describing the dissipative processes of the system.(3)As ontological particles vary throughout the reductionist hierarchy, it must be concluded that even if the phenomenological behaviors are similar, the entropies are not the same. Hence, we need to be very careful when describing such entropies, in particular ensuring that a clear definition is given and that the basic conceptual understanding used as described above is clearly described.(4)To each entropy type, a new name of classification should be given. Together with this, a consistent set of internal relations and constraints need to be described, and proper reference levels ought to be harmonized with respect to both types of entropies. Also, the organizational level of the respective system has to be fixed and delineated.(5)Two thermodynamic extremum principles are often used, namely either a maximum or a minimum entropy applied to biological systems. These views, which seem to be totally contradictory at first glance, stem from the differentiated starting points of structural versus functional entropy. Are we looking statically at the information-related distribution patterns of parts or are we observing the production of heat and wasted energy in systems of energy and matter flows? Both approaches also utilize different time scales, and the outcome of an undisturbed development can be extremely different.(6)At first glance, these results seem incompatible but introducing clearer definitions, as mentioned in #2, will be of great benefit. Through the clarification of the distinction between imposing a state view and a process view upon the systems, it could be demonstrated that it is possible to give a proposal for an initial merging of the extremes. What exactly happens is most likely not a maximization or minimization of a resulting state. Rather, it is likely to be a compromise, and an optimal stable state emerging from the internal and external constraints imposed on the respective system level.

As a result of these discussions, we can in fact ask for more terminological strictness for a higher accuracy in describing the starting points of argumentations and more detailed characteristics of the utilized entropy type, including denominations of the investigated scales. In the forthcoming steps, the interesting task of integrating the approaches within a holistic model, which is connecting structures and functions, can be developed further. From an ecological viewpoint, such a unified approach is urgently needed and extremely relevant if we observe the extreme entropy production of our societies and economies, which are now modifying the climatic constraints of our living conditions and the entropy of several spatial and informational patterns.

## Figures and Tables

**Figure 1 entropy-25-01288-f001:**
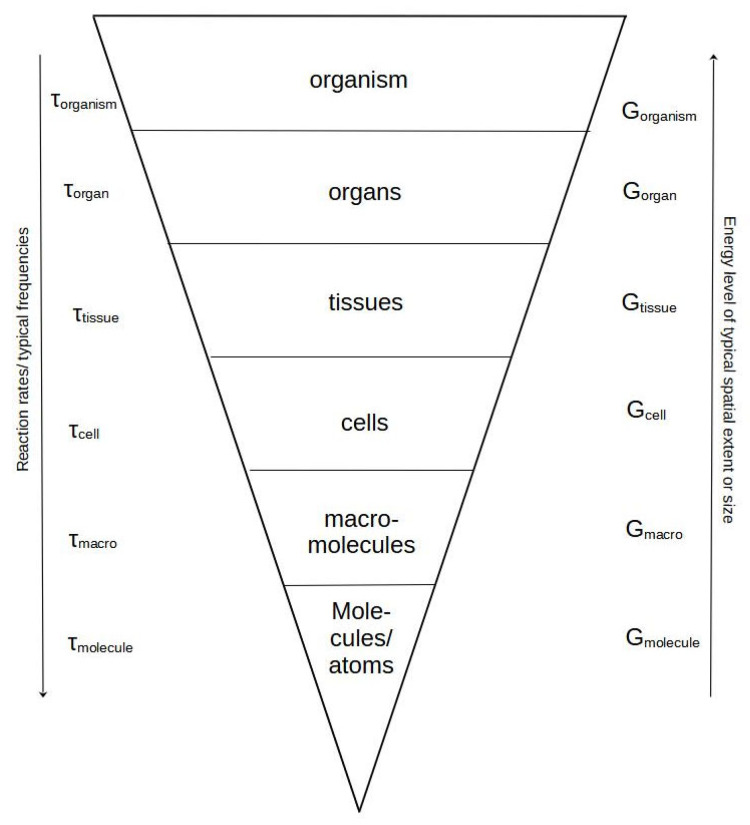
An illustration of the organism as a spatially organized hierarchy of sequentially enclosed systems from the level of molecules of various sizes through cells and their organelles via collections hereof in tissues and organs to the final organism state. According to conventional hierarchy, we find the fastest time scales and reaction rates (symbolized by the downward arrow) at the lower level (lower side of the cone). When moving upwards in size, we move in the direction of slower rates (longer physiological time scales symbolized by an increase in τ values for each hierarchical level) and usually also larger sizes and special extent. As the upper levels must include all lower levels, they are believed also to contain more structure—expressed in this case by available or Gibbs free energy (G).

**Figure 2 entropy-25-01288-f002:**
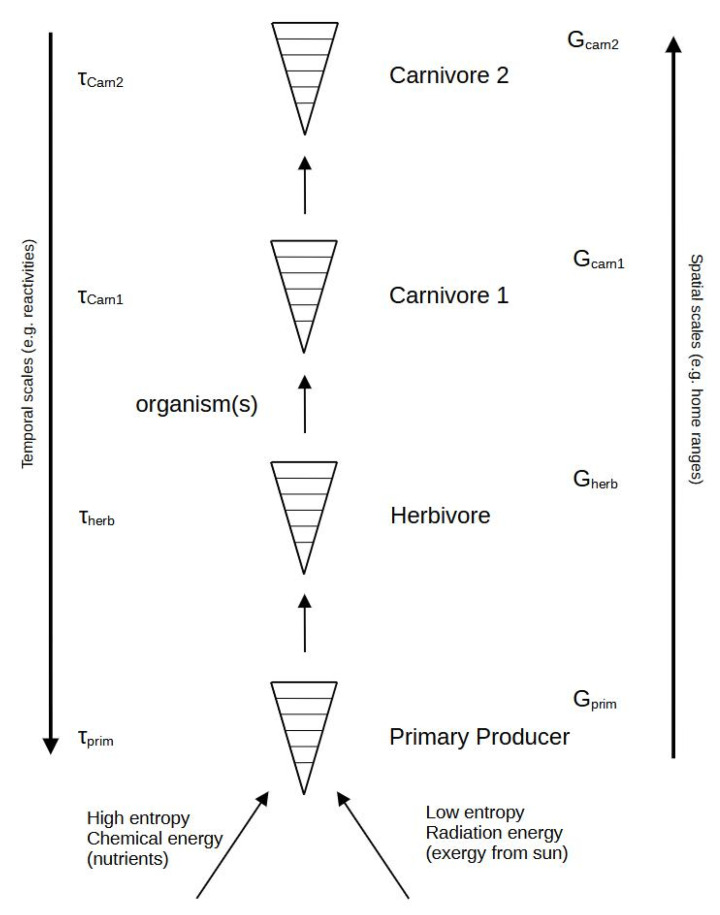
Organisms are now arranged in an ordinary food chain, demonstrating the importance of two quite different entropy value inputs at the basis of the primary producers. Without the capacity of this component to capture exergy from the solar radiation by photosynthesis, the food chain would not exist. The time scale values (τ) are believed to differ in such a way that higher-level organisms also have longer time scales and higher complexity, as reflected by their amounts of available (G). The latter can be highly questioned, as reflected in the text.

**Figure 3 entropy-25-01288-f003:**
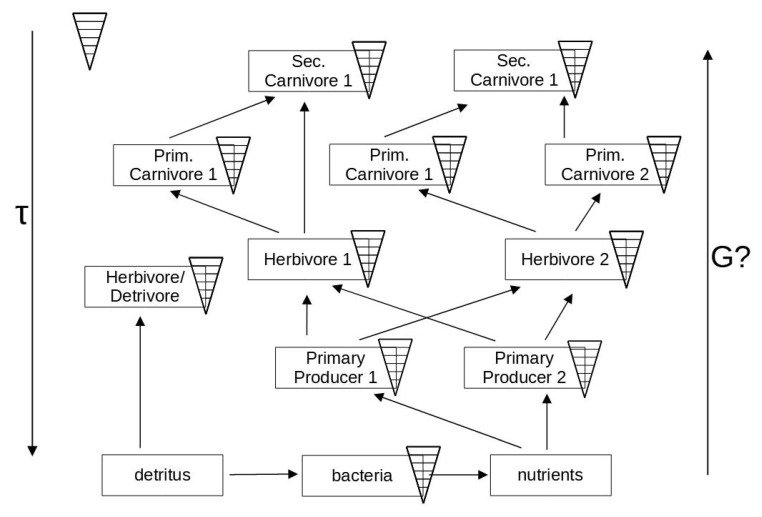
Diagram showing an ecosystem network as a set of types of organisms belonging to the same functional groups in Figure 2. The network has been expanded to allow for the recycling of the necessary nutrients within the system, which is not necessarily dependent on allochthonous matter or chemicals supplied from the outside. In principle, the system on its largest scale could be reduced to a closed system (like the Earth). The time scale of the system as a whole is determined by the different organisms together. The energetic values (G’s) cannot be distinguished in practice, for instance by calorimetric measurements.

**Figure 4 entropy-25-01288-f004:**
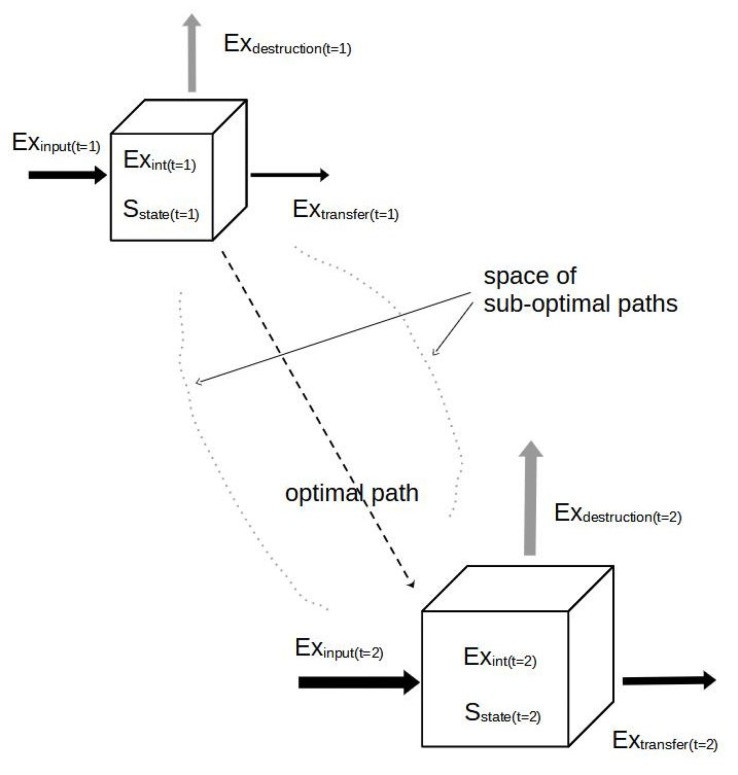
A biological open system develops through time (from t = 1 to t = 2) to increase in size, as well as the quantity and quality of processes. The figure is an attempt to visualize the situation in the discussion of using various extremum principles from thermodynamics using Lotka’s original statements as a starting point. The larger structure at t = 2 will import, store, dissipate, and export more available energy than in the situation at t = 1. The resulting structure at t = 2 can vary in size due to the path dependency of the processes necessary to bring it to the new state, but the structure doing this in an optimal manner will always win the competition, i.e., be selected and, therefore, result in the largest possible structure in accordance with prevailing constraints.

**Table 1 entropy-25-01288-t001:** List of features of life as emergent properties as found in the current literature (reproduced and amended from [83]).

Identical self-replication	Metabolic efficiency
Genetic variation	Stabilization of structure and function
Self-regulation	Reactivity
Ability of evolution	Functional processing of information
Hierarchical organization	Internal chemical steady states
Growth and development	Variability
Autocatalysis	Centripetality

**Table 2 entropy-25-01288-t002:** Hierarchical levels in biology where configurational arrangements may be interpreted as entropy or often an indication of complexity.

Level	Particles/Ontological Units	Entropy State Interpretation Examples	Entropy Production Interpretation Examples
Genome	Nucleic acids in RNA/DNA	Calculations based on frequencies of A, U, G, C or A, T, G, C	The energetic cost of maintaining this internal library is, in general, not questioned
Protein	Amino acids	Calculations based on the conversion of triplet into codes for 20 amino acids	Configuration of proteins is, in general, believed to be in accordance with a minimum free energy
Cellular	Number of organelles	Compartmentalization potentially increases entropy but also represents a necessary separation of processes	The metabolism serves the purpose of delivering the needed components through energy-costing processes Entropy formation is affected by disease and other malignant situations
Tissues	Number of cells and cell types	The separation of clusters of cells into various tissues again potentially represents an entropy increase	The separation is believed to optimize efficiency
Organs	Number of cells and cell types	Same as above but at a higher level of integration	Same as above
Organism	Organs	The organs become embedded in a boundary representing a closure—making an organism	Clear empirical results indicate that efficiency increases throughout the process of epigenesis and aging until a senescent stage
			*Physical embodiment ends somewhere near here*
			*Above this level, a constructionist approach is needed*
Population	Groups of organisms	Realization of the basic features of living systems (reproduction, evolution)	Efficiency seems to be the result of interaction and communication
Community	Assemblages of interacting populations	Groups of populations found together under the same geographical conditions Organismic inter-relations (symbiosis, parasitism)	Seems to ensure buffer capacity, resilience, or stability in function and structure
Ecosystem	Trophic levels/network/pools	Biotic–abiotic interrelations	Realization of cycling relations, ecosystem respiration, and climate regulation
Landscape/ Region	Ecosystems within a confined geographical area	Many ecosystems over a relatively wide range of climatic conditions	Results seem to have great uncertainty due to dependency on graining—coarse vs. fine does not necessarily give the same results Flows of water, energy, nutrients, and information over greater regions
Biosphere	The sum of ecosystems or biomes on Earth	Earth as one holistic socio-environmental system	Climate dynamics, global change processes, pollution, eutrophication, desertification, etc.

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
