# Peer review of "The Entropy of Entropy: Are We Talking about the Same Thing?"

_entropy, 2023, doi:10.3390/e25091288_

Round 1

Reviewer 1 Report

I did not find any flaws in the paper, and think that it could be published as is.

Author Response

C: I did not find any flaws in the paper, and think that it could be published as is.

R: no changes made as a result of the comments from this reviewer

Reviewer 2 Report

See uploaded file.

Already answered above.

Author Response

Unfortunately, some reformatting has happened so that the line numbers given are not consistent between present manuscript and reviewers version. I have not been able to identify all positions - but most :-)

This reviewer recommends acceptance of the paper with minor alterations.

Specific Comments:

Comment: Line 83: Swenson 1997 as well
Response: reference added to the list

Comment: Lines 130-140: The idea that living, complicated systems produce more entropy
than the physical system without them was discussed by Ulanowicz and Hannon 1987. Life
and the production of entropy. Proceedings of the Royal Society, London Series B 232:181-
192. See also Lines 1089 –
1098.
Response: Reference to this work added where appropriate

Comment: Line 150: Onsager introduced linear relationships into the discussion. See also
line 871.
Response: Onsager's contribution has been acknowledged at several places now

Comment: Line 201: the reference levels are required by the third law of thermodynamics.
Response: This place has not been precisely identified - but a section discussing the
importance of reference levels is found elsewhere in the paper

Comment: Lines 214 – 234: The Lotka principle deals with positivist drivers. Entropy is not
such an entity. Greater power does create entropy at a greater rate, however. This is part of a
Hegelian dialectic (see below).
Response: The dialectic view is introduced at the end of the paper. While we agree fully, just
the term meanwhile still seem quite controversial among natural scientist due to implicit
reference to Engels' Nature Dialectics - therefore it was chosen to introduce the term as late
as possible
Comment: Line 225: Thermodynamics is the invention of engineers, not physicists!
Physicists were dissatisfied with the implications of the second law and created their own
(sanitized) version of thermodynamics predicated upon statistical mechanics.
Response: We believe this gap is also identified several places - but again the purpose is
unification through understanding of respective views rather than splitting up even further.
Comment: Lines 259 – 260: Chemically bound energies are positivist properties; destroyed
dissipative energies are lacunae, and hence, ontologically different.

Response: Agree fully - an attempt to make a distinction has been introduced.

Comment: Line 299: heat *loss* -- an absence.
Response: Not really understood - heat in physiology can be lost or gained over the boundary
of an organism - this what we refer to.
Comment: Line 306: Yes, the systems are out of equilibrium, but they nevertheless reach a
balance.
Response: Point taken - the point is made elsewhere.
Comment: Lines 336 – 342: Entropy, the step-child of physics.
Response: Agree, - sometimes it really is an orphan of many sciences - hence this paper.
Comment: Line 395: From a network perspective, too great an efficiency can become a
liability.
Response: This point entered the paper elsewhere - as a discussion of avoiding chaotic
regimes.

Comment: Line 406: (dynamic) equilibrium = balance
Response: The balanced view was added.

Comment: Line 431: Despite Schroedinger’s obvious genius, “negentropy” is an oxymoron.
The negative of zero is still zero. He meant, of course, mutual constraint.
Response: So right - oxymoron is the right word for this.

Comment: Line 457: “entropy free energy” confuses matters. Free energy or exergy would
suffice.
Response: Right free energy - it is now.

Comment: Line 466: “entropy as information” is a contradiction. It stems from Weaver’s
misattribution.
Response: This misfortune (and oxymoron as well) has now been addressed.

Comment: Lines 519 -521: Physicists remain wedded to ideal gases. Alas!
Response: Some but not all - we attempt address both sides to create convergence in
discussions.

Comment: Line 539: Yes, the “driving factor” was *mis-named* negentropy. Drivers are
either physical forces or agencies.
Response: Attempt of elaboration made.

Comment: Lines 597 – 622: It is often neglected that states and rates have different physical
dimensions. Rates include time; states do not. For an engineer that is a *major* difference.
Response: Distinction was accentuated further in text.

Comment: Line 607: optimized or balanced?
Response: Well, probably both - attempt of amendment - but not sure if it is the right location.

Comment: Lines 622 – 634: John Harte is a leader in the MAXENT in ecology and should be
referenced. See also lines 700 – 713.
Response: Harte is now added to the references here.

Comment: Lines 748 – 762: Maybe add mutual information and “conditional” entropy?
Response: Right - they have both been added.

Comment: Line 786: At this level of description, may we forget about “mechanisms” and talk
about “corresponding phenomenological principle?
Response: Agree - principle is a better word - while writing couldn't find a replacement for
mechanism.

Comment: Lines 790 – 804: The bottom line is that physical reductionism is not always
possible. A good point!
Response: Thanks.

Comment: Line 811: I don’t think Elsasser said ALL states are unique, but rather MANY are.
Response: At least we said it somewhere else in references - but the sentence has been
modified according to criticism

Comment: Lines 817 – 849: good discussion. “ontic openness” = indeterminacy, but maybe it
is a wise substitution to avoid flak from determinist physicists.

Response: Well, ontic openness is the feed of both newness and constrains and as such should
not be ignored.

Comment: Line 859: Mae Wan Ho et al. in Bioenerg. 41, 81–91 [1996] maintained that
living systems are in quasi-equilibrium.
Response: We could not identify the exact location (modified line numbers), so reference was
added around section on organism level.

Comment: Lines 897 – 898: How about centripetality? It is such an important attribute of all
life, but nobody includes it on their list of essential features!
Response: autocatalysis and centripetality added to list.

Comment: Line 971 – 973: See Zorach Complexity 8(3):68-76 [2003].
Response: Thanks for reminding us of the paper.
Zorach, A. C., & Ulanowicz, R. E. (2003). Quantifying the complexity of flow networks: how many
roles are there?. Complexity, 8(3), 68-76.

Comment: Line 1023, Figure 3: Such trees can be consolidated into trophic chains. See pp.
549-560. In Complex Ecology: The part-whole relation in ecosystems [1995]. See also lines
1297 – 1310.
Response: Agree - but we believe this is complicated enough - as elaboration would lead
further into network (thermodynamic) analysis.

Comment: Line 1077 – 1078: “restricting their degrees if freedom” = creating information
(AMI).
Response: Attempt of amendment made.

Comment: Lines 1137 – 1138: See Mae-Wan Ho on organisms as near-equilibrium systems.
Response: Added to organism discussion on this.

Comment: Lines 1160 – 1181: See Stuart Kauffman’s narrative on the transition from
plankton to periphyton. Predicated on J. Gavis, J. Mar. Res. 34:161–179 [1976].
Response: We are not sure exactly what the point here is - and are not able to identify
Kauffman's statements - Kauffmann was added under "edge of chaos" section.

Comment: Lines 1207 – 1212: See also Ecological Modelling 220:1886-1892 [2009]. As
well as 1369 –1372, and 1665 – 1668.
Response: Thanks for the interesting notice.

Comment: Lines 1382 – 1384: The property, “ascendency” was created as a homolog of the
Gibbs Free Energy.
Response: Amended

Comment: Lines 1395 – 1398: Mann et al. in Mathematical Models in Biological
Oceanography UNESCO 1980, decried the absence of flows in ecological analysis. See also
lines 1456 -1457 and 1454 – 1462.
Response: Reference not available and location not identified - but the focus on entropy as
state descriptions is already discussed in the paper.

Comment: Line 1405 – 1406: Entropy of flow networks can be quantified by information
theory applied to flow networks.
Response: Right - but we tend to avoid the entropy/information discussion - even for
networks.

Comment: Lines 1430 – 1432: Patten is correct that the input environ is a driver, but the
ultimate driver is centripetality.
Response: We have made an attempt to distinguish between the two types of drivers as
external and internal respectively.

Comment Line 1479: entropy = That which does not exist or no longer exists. Such a
definition clears up a lot of ambiguity!
Response: Interesting aspect.

Comment: Lines 1517 – 1521: Differences in physical dimensions are extremely important
(to engineers).
Response: The distinction between describing the system as state and/or flows has been
accentuated and now enters summarizing conclusions

Comment: Lines 1554-1555: ”negative change in entropy” = creation of constraints.
Response: Redundant.

Comment: Lines 1468 – 1474: Yes, see Ecology Letters 17:127–136 [2014].
Response: Reference added.

Comment: Lines 1606 – 1607: That depends. If the system is over-balanced with AMI, it
might not.
Response: Comments around this entered with Kauffman.

Comment: Lines 1639 – 1649: See Michael Conrad. 2012. Adaptability: The significance of variability from molecule to ecosystem
Response: Locations not identified exactly - so exact character of issue not identified.
Conrad, M. (Ed.). (2012). Adaptability: The significance of variability from molecule to ecosystem.
Springer Science & Business Media.

Comment: Lines 1648 – 1649: Nor need it do so!
Response: Unclear for us because of uncertain location.

Reviewer 3 Report

I appreciate the tough work made by the authors to try to clarify the concept of entropy; however, despite the paper is very comprehensive it does not add the necessary and needed simplification into this ''entropy debate''. In my mind the definition of entropy is one, at least conceptually. The work of Amos Maritan and John Harte should be followed where a conceptual and analytical unifying clarification was made about entropy. Authors should also check this paper https://doi.org/10.1126/sciadv.1701088 where a broader information-theoretic perspective has been derived and applied inspired by the ecosystem's entropy foundations of Maritan and Harte. In any event, I agree that there is  a lot of confusion but this confusion is driven by people from different disciplines that believe entropy to be different in different fields or just because the analytics is formulated differently.  The concept is however the same: what may be different is the space-time reference scale.  like the paper but I feel it is very long and should provide some suggestion to resolve this ''Gordian entropy knot''. It reads as a review more than a conceptual paper where solutions are found. I would be very glad to read what the conclusions are for the authors. 

None

Author Response

We appreciate the comments from the reviewer very much as he/she clearly represents a direction of "entropic principle" application to which we should like to build a bridge.

Comment: I appreciate the tough work made by the authors to try to clarify the concept of entropy; however, despite the paper is very comprehensive it does not add the necessary and needed simplification into this ''entropy debate''. In my mind the definition of entropy is one, at least conceptually.

Response: We believe to have given several clear proposal for change of semantics in the area which should serve to clarify communication in the area. We agree that this is not necessarily a simplification - but believe that accentuation on definitions and clear statements is the only way to improve the works in the area towards, - perhaps as final reconciliation or resolution of the debate.

We agree on the latter statement - but while the behavior of the equation for "entropy" might not change phenomenologically over the various hierarchical levels of biology - the huge difference in ontological components does not allow us to conclude that the "entropies" are all the same. Some amendments have been made in an attempt to accentuate this point.

 Comment: The work of Amos Maritan and John Harte should be followed where a conceptual and analytical unifying clarification was made about entropy.

Response: We have studied the respective papers and added their references where they could enter the text in a proper manner.

Comment: Authors should also check this paper https://doi.org/10.1126/sciadv.1701088 where a broader information-theoretic perspective has been derived and applied inspired by the ecosystem's entropy foundations of Maritan and Harte.

Response: Text has been checked and reference has been added - but it should be noted that the calculations in this paper mainly deal with qualitative relations.

Comment: In any event, I agree that there is  a lot of confusion but this confusion is driven by people from different disciplines that believe entropy to be different in different fields or just because the analytics is formulated differently.  The concept is however the same: what may be different is the space-time reference scale.  

Response: We think that scale is not the nly field of problems / difference; see above.

 Comment: like the paper but I feel it is very long and should provide some suggestion to resolve this ''Gordian entropy knot''.

Response: Several sections were already dedicated to this - in particular the whole last section #10 was sketching a first of attempt of resolution of the confusion debate. We have also been asthonished about the length of the paper but we think that the whole text is necessary.

A summary of the demanded suggestions made throughout the paper have now been summarized at the end, and the objectives have been reformulated in the end of Chapter 1.

Comment: It reads as a review more than a conceptual paper where solutions are found. I would be very glad to read what the conclusions are for the authors. 

Response: the paper was submitted as a review - but deviating from normal reviews as it works on unifying concepts rather than just referring to contributors.

Reviewer 4 Report

I have really appreciated this work,
the topic is interesting, the study is
well-written and well-presented.

Thus I have only minor remarks.

- Keywords: in alphabetical order, please

- all the references in the main text should be numbered
in order of appearance and indicated by a numeral or numerals in square brackets, as required by the journal

- Equation 1: the term "l" in not defined in the text

- Equation 2:  the number of possible microstates is now indicated by "W" while there was a greek letter in Equation 1. Choose which to use

- Equation 2: which base for the log?

- Equation 3: which base for the log?

- Equation 4: place properly "pi" that should be the subscript of the summation. Otherwise it seems that you are summing pi*ln(pi)

- Equation 4: why the "log" of the previous equations has now become a natural logarithm ln? To change from log to ln a transformation through a constant is required

- Equation 7: the term on the left is not defined

- Equation 7: this equation is exactly equal to Equation 6 as Itherm = Smax - Sstate

- avoid bold text outside titles (rows 1485, 1510, 1517)

- why Table 1 is partly in the main text (row 896)
and in part after the Acknowledgments (row 1682)?

Author Response

C(omment): I have really appreciated this work, the topic is interesting, the study is well-written and well-presented.

Thus I have only minor remarks.

R(esponse): We are happy that this reviewer appreciated our efforts

C: - Keywords: in alphabetical order, please

R: this is fixed

C: - all the references in the main text should be numbered in order of appearance and indicated by a numeral or numerals in square brackets, as required by the journal

R: Right, - is fixed after inserting new references

C: - Equation 1: the term "l" in not defined in the text

R: Fixed with remark inserted. In his original text Boltzmann used a letter (el) to indicate that he expected a logarithmic dependency - not considering which one exactly

C: - Equation 2:  the number of possible microstates is now indicated by "W" while there was a greek letter in Equation 1. Choose which to use

R: The historical development of this is now indicated in text

C: - Equation 2: which base for the log?

R: all logarithmic dependencies have now been changed to natural logarithms

C: - Equation 3: which base for the log?

R: - same as previous

C: - Equation 4: place properly "pi" that should be the subscript of the summation. Otherwise it seems that you are summing pi*ln(pi)

R: fixed - summation is now over i

C: - Equation 4: why the "log" of the previous equations has now become a natural logarithm ln? To change from log to ln a transformation through a constant is required

R: same response as above - and text about conversion is inserted

C:- Equation 7: the term on the left is not defined

R: fixed - as the left side refers to exergy

C: - Equation 7: this equation is exactly equal to Equation 6 as Itherm = Smax - Sstate

R: Smax has now been replaced with Sref - to indicated that a reference state other than maximum is most likely a more proper way of estimating the exergy of a biological system

C: - avoid bold text outside titles (rows 1485, 1510, 1517)

R: Changed to italics

C: - why Table 1 is partly in the main text (row 896)
and in part after the Acknowledgments (row 1682)?

R: There are two tables - is corrected and should be clear in the revised version